# *JUN* upregulation drives aberrant transposable element mobilization, associated innate immune response, and impaired neurogenesis in Alzheimer's disease

Chiara Scopa [1,2] ✉, Samantha M. Barnada [1], Maria E. Cicardi[2], Mo Singer[2], Davide Trotti [2,4] ✉ & Marco Trizzino [1,3,4] ✉

Adult neurogenic decline, inflammation, and neurodegeneration are phenotypic hallmarks of Alzheimer's disease (AD). Mobilization of transposable elements (TEs) in heterochromatic regions was recently reported in AD, but the underlying mechanisms are still underappreciated. Combining functional genomics with the differentiation of familial and sporadic AD patient derived-iPSCs into hippocampal progenitors, CA3 neurons, and cerebral organoids, we found that the upregulation of the AP-1 subunit, c-Jun, triggers decondensation of genomic regions containing TEs. This leads to the cytoplasmic accumulation of HERVK-derived RNA-DNA hybrids, the activation of the cGAS-STING cascade, and increased levels of cleaved caspase-3, suggesting the initiation of programmed cell death in AD progenitors and neurons. Notably, inhibiting c-Jun effectively blocks all these downstream molecular processes and rescues neuronal death and the impaired neurogenesis phenotype in AD progenitors. Our findings open new avenues for identifying therapeutic strategies and biomarkers to counteract disease progression and diagnose AD in the early, pre-symptomatic stages.

Alzheimer's disease (AD), which is the most common form of dementia, is an age-related neurodegenerative disorder characterized by progressive memory loss and a decline of cognitive function[1-3]. The disease is classified as familial AD, associated with mutations in three major genes (*APP*, *PSEN1*, *PSEN2*), and sporadic AD which arises without a genetic mutation[4]. The histopathological hallmarks of AD include the accumulation of extracellular Amyloid beta (Aβ) plaques and intracellular neurofibrillary TAU tangles (NFTs)[5] in the brain. The hippocampus is one of the first regions of the brain to accumulate these pathological features during the early stages of AD[6]. Additionally, the subgranular zone of the dentate gyrus in the hippocampus is a human neurogenic niche[7-9], which harbors neural stem cells and controls cell fate determination[10]. In line with this, AD shows impaired adult hippocampal neurogenesis which is also a common event among various neurodegenerative disorders[11-14].

[1]Department of Biochemistry and Molecular Biology, Thomas Jefferson University, Philadelphia, PA, USA. [2]Jefferson Weinberg ALS Center, Vickie and Jack Farber Institute for Neuroscience, Department of Neuroscience, Thomas Jefferson University, Philadelphia, PA, USA. [3]Department of Life Sciences, Imperial College London, London, UK. [4]These authors jointly supervised this work: Davide Trotti, Marco Trizzino. ✉e-mail: chiara.scopa@jefferson.edu; Davide.Trotti@jefferson.edu; m.trizzino@imperial.ac.uk

Several studies have reported alterations in hippocampal neurogenesis in transgenic animal models of AD[13] and a decline of adult neurogenesis in AD patients[15,16]. Notably, these alterations occur in the early stages of the disease[13], suggesting that defects in neurogenesis can trigger the onset of AD clinical phenotypes. Consequently, impaired neurogenesis likely accelerates and facilitates this neurodegenerative progression[17]. Moreover, these studies highlight the link between the key hallmarks of AD (i.e., TAU and Aβ) and neurogenesis. Recent studies also indicate that TAU plays a crucial role in the microtubule dynamics required for axonal outgrowth and that the hyperphosphorylation of TAU impairs hippocampal neurogenesis[17–19]. Emerging evidence suggests that intracellular accumulation of Aβ also negatively impacts neural precursor cell (NPC) proliferation and hippocampal neurogenesis[20–22].

Neural stem cell fate determination and neurogenesis are regulated by MAP kinases[23–26]. Several studies have unveiled a compelling link between MAPK signaling and AD pathogenesis, demonstrating that the c-Jun-amino-terminal Kinase (JNK) pathway is involved in Aβ-induced neurodegeneration[27–30] and in the hyperphosphorylation of TAU, thus contributing to the formation of the NFTs[31–33].

Importantly, c-Jun is the downstream effector of the JNK pathway. Phosphorylated-c-Jun is a fundamental member of the AP-1 family of transcription factors, functioning as either a homodimer (c-Jun/c-Jun) or a heterodimer (c-Jun/c-Fos, c-Jun/ATF2, c-Jun/MAF)[34]. Among various other functions, AP-1 modulates cell death signaling[31,32] and promotes the transcription of a series of pro-apoptotic factors, such as TNF-α, FAS-L, c-MYC, and ATF3, which induce cell death via apoptosis[31–33]. Recent studies suggest that AP-1 can act as a pioneer factor by binding condensed nucleosomes and recruiting chromatin remodelers, such as the BAF complex, to elicit chromatin accessibility[35–38]. Notably, JUN (encoding for c-Jun) is upregulated in neurodegenerative diseases, including AD[32,36]. Nonetheless, the explicit link between aberrant c-Jun activity and the associated neurodegenerative outcomes has yet to be explored in depth.

Finally, there is mounting evidence indicating a role for transposable elements (TEs) in the molecular pathogenesis of AD. More specifically, this disease is characterized by aberrant de-repression and mobilization of TEs found in regions of repressed chromatin, particularly retrotransposons belonging to the long interspersed nuclear element (LINE) and long terminal repeat (LTR) families[37–44]. Yet, the mechanisms leading to TE de-repression and the functional consequences of this phenomenon in AD pathogenesis are understudied, especially in humans. Recent studies showed that overexpression of TAU alone in aging *Drosophila* brains is sufficient to increase the expression of retrotransposons, mostly belonging to the LINE and ERV groups[38,41]. However, the mechanism linking tauopathies to chromatin relaxation and TE mobilization remains unexplored.

In this study, we differentiated familial and sporadic AD patient-derived induced pluripotent stem cells (iPSCs) into hippocampal progenitors, CA3 neurons, and cerebral organoids. We demonstrated that c-Jun is the upstream regulator of the transcriptional network altered in AD hippocampal progenitors and that the aberrant upregulation of JUN leads to the de-repression and mobilization of hundreds of TEs. Moreover, we found that aberrant TE mobilization induces a cytoplasmic accumulation of RNA–DNA hybrids, which elicits the activation of the cGAS–STING pathway and caspase-3, suggesting the initiation of programmed cell death. Inhibiting c-Jun phosphorylation/activation blocks this pathological axis in AD progenitors by maintaining TE repression, ultimately preventing the activation of the downstream pathogenic cascade.

## Results
### Human iPSC-derived model of human neurogenesis in familial AD
To investigate the role of c-Jun in the onset of AD, we used familial AD cell lines. In detail, we derived hippocampal precursor cells (hpNPCs)

and CA3 neurons from these AD patient-derived lines and controlled human iPSCs. First, we confirmed the pluripotency and the expression of JUN in two controls (CTRL1 and CTRL2) and two familial AD lines (FAD1 and FAD2). The CTRL and FAD lines were both sex and age-matched. The FAD1 cell line contains an *APP* gene duplication and the FAD2 line has a heterozygous missense mutation in *PSEN2* (*PSEN2*:p.Asn141Ile); both genetic variants are associated with familial AD.

Immunofluorescence staining demonstrated there is no significant difference in NANOG and OCT4 expression between CTRL and AD lines, indicating that all these iPSC lines are equally pluripotent (Supplementary Fig. 1a). Additionally, CTRL and AD lines have roughly the same percentage of cells that express c-Jun, but FAD iPSCs display significantly higher protein expression (Supplementary Fig. 1b).

We first differentiated the four iPSC lines (CTRL1, CTRL2, FAD1, and FAD2) to hippocampal NPCs (hpNPCs) using a previously published protocol (Fig. 1a)[45]. We next quantified the expression of established markers for different hippocampal neural precursor stages and assessed the composition of our obtained hpNPC population (Fig. 1b). *NESTIN* defines early precursors, *TBR2* and *FOXG1* define intermediate progenitors, *PROX1* late progenitors, while *DCX* defines neuroblasts[46] (Fig. 1b). We observed a considerable reduction in the expression of the early neural stem cell marker (*NESTIN*) and an increase in the expression of the intermediate progenitor markers (*TBR2* and *FOXG1*) in the FAD hpNPC population (Fig. 1b). This signature suggests impaired neurogenesis in the FAD hpNPCs. This result was also confirmed through immunofluorescence (Fig. 1c, d). Moreover, the FAD hpNPCs not only had less DCX-positive cells (Fig. 1c, d) but also showed reduced expression of DCX compared to CTRL hpNPCs, indicating that the FAD lines did not properly differentiate into neuroblasts (Fig. 1b).

### *JUN* modulates the transcriptional network dysregulated in familial AD hippocampal neural progenitors
To further investigate the differences between CTRL and FAD hpNPCs, we performed RNA extraction followed by sequencing (RNA-seq) to characterize their distinct transcriptomes. After 20 days in the proliferation medium, we collected the cells to perform RNA-seq. This analysis identified 1973 differentially expressed genes, 718 of which (36.4%) were downregulated, and 1255 (63.6%) were upregulated in the FAD progenitors (FDR < 5%; $\log_2$(FC) ±1.5; Fig. 2a). In line with our RT-qPCR and immunofluorescence data (Fig. 1), the early progenitor markers, *NESTIN* and *PAX6*, were downregulated in FAD progenitors confirming our impaired neurogenesis observation.

Several studies have demonstrated a critical role for WNT signaling in the pathogenesis of AD[47–53] and in regulating adult hippocampal neurogenesis[54–61]. Accordingly, the expression of several genes involved in both the canonical (*DKK3*, *WNT7A*, *SFRP4*, *WNT2*) and non-canonical (*WNT5A*, *RORA*, *RAC2*) WNT signaling pathways was upregulated in FAD hpNPCs (Fig. 2a). Moreover, *DKK1* was more expressed in the FAD lines relative to the CTRLs (Supplementary Table 1). DKK1 is an antagonist of the canonical WNT signaling pathway[62], leading to the activation of the WNT/JNK pathway and ultimately resulting in increased phosphorylation (i.e., activation) of c-Jun[63]. Furthermore, the expression of *JUN* itself was significantly upregulated in the FAD lines ($\log_2$(FC) = 0.6576124; *P*-value = 0.00904464; Fig. 2a). Consistent with this, the activation of the WNT/JNK pathway is suggested to play a role in Aβ oligomer neurotoxicity[49,64].

We employed the WEB-based GEne SeT AnaLysis Toolkit (WebGestalt)[65] to identify pathways associated with 1973 differentially expressed genes. This analysis revealed that these genes are associated with inflammation, neurogenesis, and neural differentiation, as well as cytoskeleton organization, apoptotic process, and the MAPK cascade (Fig. 2b). Notably, ingenuity pathway analysis (Qiagen) identified c-Jun

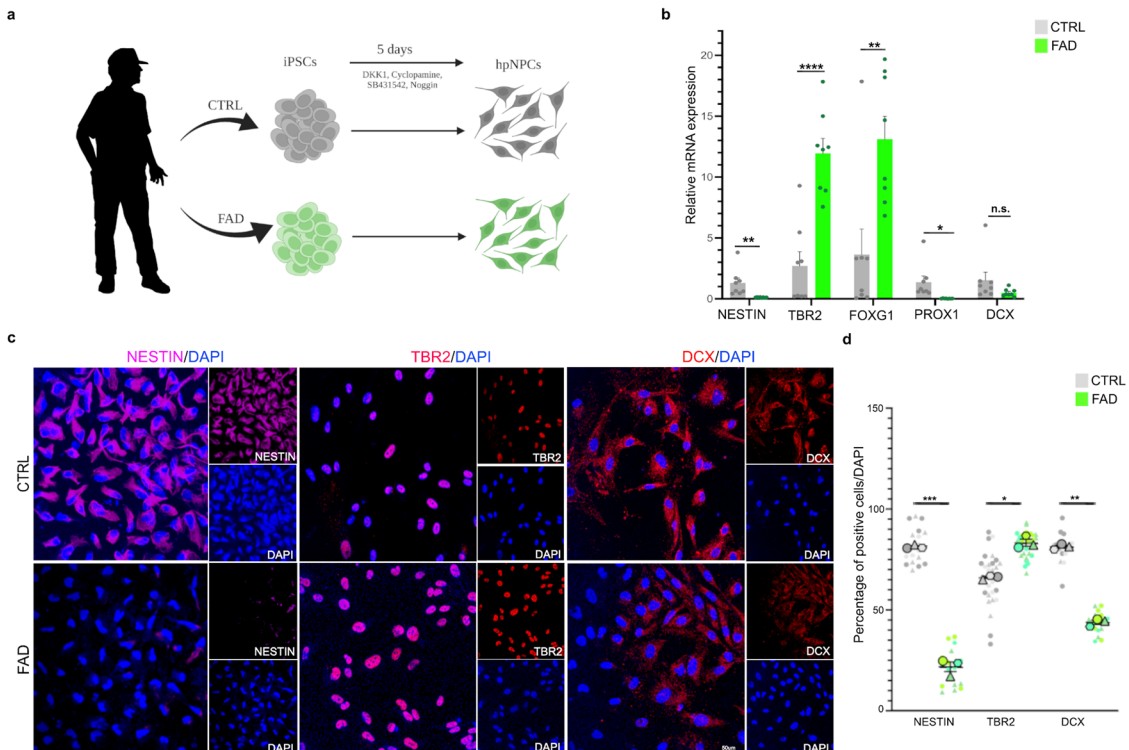

**Fig. 1 | FAD iPSC-derived hippocampal neural progenitors display impaired neurogenesis. a** Scheme of the protocol for hippocampal neural progenitor cell (hpNPC) differentiation (made with BioRender.com). iPSCs were derived from the skin fibroblasts of two patients with familial Alzheimer's disease (FAD1 and FAD2) and two healthy controls (CTRL1 and CTRL2), and differentiated into hpNPCs after 5 days in induction media. The hpNPCs were maintained in proliferation media post-induction. **b**−**d** qPCR and immunofluorescence for markers of different stages of hpNPC populations. NESTIN early precursors, TBR2/FOXG1 intermediate progenitors, PROX1 late progenitors, DCX neuroblasts. **b** The qPCR shows a neurogenic defect in FAD progenitors, with enrichment for TBR2-positive intermediate progenitors as confirmed in the immunofluorescence and relative quantification for hpNPC population markers. Each dot in the bar plot represents an experiment (*n* = 8 independent experiments). A two-sided *t*-test was used for the comparison between the two groups. A value of *P* < 0.05 was considered significant; *\**P* < 0.05

(*PROX1 P* = 0.018119); *\*\**P* < 0.01 (*NESTIN P* = 0.009623 and *FOXG1 P* = 0.004541); *\*\*\**P* < 0.001 (*TBR2 P* = 0.0000075); n.s. not significant (*DCX P* = 0.148171). Error bars report the standard error. **c**, **d** immunofluorescence and its quantification. IF Scale bar 50 μm, 40× magnification. DAPI staining on nuclei in blue. Each IF was replicated three times with similar results (*n* = 3 independent experiments). The IF data are quantified using a Superplot, which concisely visualizes individual data points and their averages. The distinct combinations of colors and shapes indicate the three independent experiments performed. Each small dot in the graph corresponds to a specific data point representing an analyzed image or cells. The larger dots represent the average values calculated from the respective data points. A repeated measures ANOVA test was used for the comparison between the two groups. A value of *P* < 0.05 was considered significant; *\**P* < 0.05 (TBR2 *P* = 0.0118); *\*\**P* < 0.01 (DCX *P* = 0.006); *\*\*\**P* < 0.001 (NESTIN *P* = 0.000571). Error bars report the standard error.

as one of the enriched upstream transcriptional regulators to all of the differentially expressed genes, suggesting that most of these genes are direct or indirect c-Jun targets (Fig. 2c). Moreover, when analyzing the enriched pathways that included the greatest number of differentially expressed genes, *JUN* was one of only 8 genes functioning in all these pathways (Fig. 2d). To test this computational prediction, we performed an immunoblot to confirm the upregulation of c-Jun and phosphorylated c-Jun in FAD hpNPCs (Fig. 2e).

These results indicate that the FAD progenitors are characterized by aberrant activation of the WNT/JNK pathway, upregulation of *JUN*, and dysregulation of its target genes.

## The WNT/JNK pathway is dysregulated in FAD CA3 hippocampal neurons

Recent AD studies have demonstrated that canonical WNT signaling is inhibited by several pathogenic mechanisms leading to neural death and synaptic plasticity impairment[51,52]. To investigate whether aberrant activation of the WNT/JNK pathway, as seen in FAD hpNPCs, was also occurring in AD neurons, we differentiated CTRL and FAD progenitors into CA3 hippocampal neurons using an established protocol (Supplementary Fig. 2a)[66]. Both fully differentiated CTRL and FAD neurons expressed the specific CA3 markers, including glutamate ionotropic

receptor kainate type subunit 4 (GRIK4) and secretagogin (SCGN; Supplementary Fig. 2b). However, the differentiation of the FAD lines resulted in a reduced number of mature CA3 neurons relative to the CTRL (51.4% in FAD; 87.2% in CTRL), despite the same percentage of late progenitors in the culture (97.4% in CTRL and 97.3% in FAD, Supplementary Fig. 2b).

We performed RNA-seq on the CA3 neurons and identified 563 differentially expressed genes between CTRL and FAD (FDR < 5%; Supplementary Fig. 2c). Approximately 80 of these genes, including *ROR2*, *WNT7A* and *WNT7B*, are involved in both the WNT/JNK pathway and the MAPK cascade. Notably, using WebGestalt, we identified the MAPK cascade as one of the top 10 differentially expressed pathways in FAD neurons (Supplementary Fig. 2d). Interestingly, MAPT is the only gene that functions in a majority of these neuronal processes (Supplementary Fig. 2e). *MAPT* encodes for TAU which is one of the key proteins associated with AD pathogenesis via the formation of NFTs[67,68]. Recent studies have reported a correlation between aberrant c-Jun activity and the formation and maturation of these NFTs[32].

In summary, we observed that the WNT/JNK pathway is dysregulated, not only in FAD hpNPCs but also in FAD CA3 hippocampal neurons, suggesting a critical role for its main effector, c-Jun, in both these cell types.

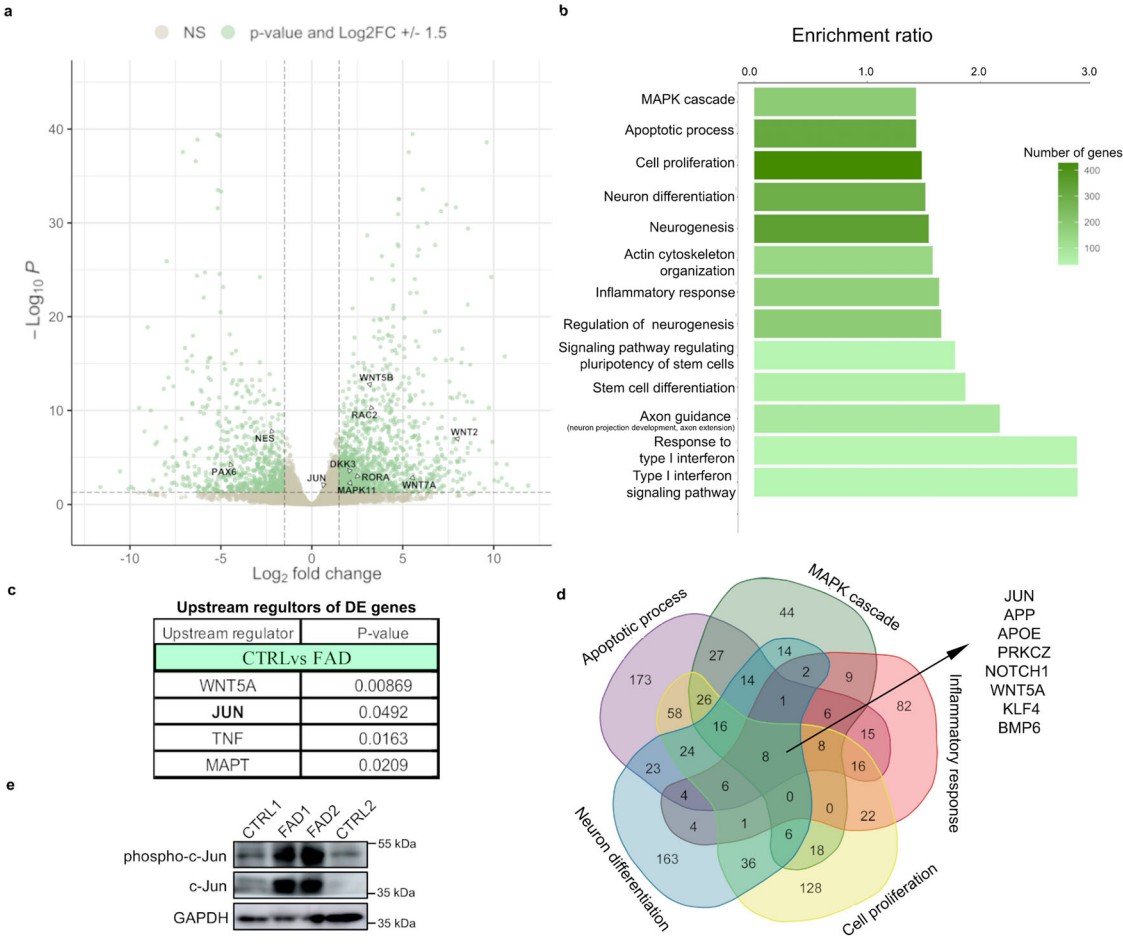

**Fig. 2 | *JUN* upregulation underlies dysregulated transcriptional networks in FAD hpNPCs. a** Volcano plot showing genes differentially expressed in FAD hpNPCs relative to CTRL hpNPCs. Labeled differentially expressed genes are involved in neurogenesis (*PAX6*, *NES*) and WNT/JNK signaling (*WNT2*, *WNT7A*, *WNT5B*, *RAC2*, *RORA*, *MAPK11*, *DKK3*). Green = differentially expressed genes passing significance thresholds *P*-value < 0.05 and log₂(fold-change) ± 1.5; Gray = not significant. A 5% false discovery rate (FDR) was used to correct for multiple testing. **b** Enriched pathways associated with 1976 differentially expressed genes in FAD hpNPCs predicted by WebGestalt. A 5% false discovery rate (FDR) was used to correct for multiple testing. **c** Top upstream regulators/transcription factors of 1976 differentially expressed genes in FAD hpNPCs, as predicted by ingenuity pathway analysis (Qiagen). A 5% false discovery rate (FDR) was used to correct for multiple testing. **d** Venn diagram showing the genes shared across all of the top five enriched pathways with the most differentially expressed genes (venn diagram made with https://bioinformatics.psb.ugent.be). **e** Immunoblot displaying the upregulation of both c-Jun and phosphorylated c-Jun across FAD and CTRL hpNPCs. The blot was replicated three times with similar results (*n* = 3 independent experiments).

## Dysregulated chromatin accessibility in FAD hippocampal progenitors

To investigate if the transcriptomic aberrations identified in the FAD lines are associated with significant differences in chromatin accessibility, we performed ATAC-seq on the CTRL and FAD progenitors, generating 150 bp long Paired-End reads.

We identified 3382 differentially accessible (DA) regions between CTRL and FAD hpNPCs (FDR < 5%; log₂(FC) ± 1.5). Of these regions, 28.8% were significantly more accessible in FAD progenitors compared to CTRL (FAD Up; Fig. 3a). By examining the nearest gene to each of the 3382 DA regions, we found that 15.1% (512) of the DA regions were located nearest to a differentially expressed gene (Fig. 3b), suggesting that there are at least 512 enhancer-gene pairs (or promoter-gene pairs) dysregulated in the FAD progenitors. Of the 512 DA regions, 95.7% were putative enhancers (distance from closest transcription start site [TSS] > 1 kb), whereas 4.3% were putative promoters (TSS distance <1 kb; Fig. 3c).

We then performed DNA-based motif analysis (MEME-ChIP) to identify any potential transcription factors underlying these changes in chromatin accessibility in the 3382 DA regions. Remarkably, the c-Jun

binding motif was the most significantly enriched in the DA regions (e-value = $2.7 \times 10^{-10}$; Fig. 3d).

## TEs are aberrantly active in FAD hippocampal progenitors

Aberrant de-repression of TEs is an emerging hallmark of AD[40,44]. In agreement, we observed that 1437 DA regions overlapped with a TE. The top-50 TE copies more accessible (i.e., active) in FAD relative to CTRL progenitors were predominantly (94%) retrotransposons (RTEs). Of these re-activated RTEs, 45.6% were LTRs, while the remaining were more or less equally distributed between LINEs (long interspersed nuclear elements) and SINEs (short interspersed nuclear elements; Fig. 3e). To further investigate the aberrant RTE mobilization in FAD hpNPCs, we looked at the distribution of TE family enrichment across the DA regions. The LTR family exhibits the highest level of aberrant accessibility in FAD lines (CTRLvsFAD) compared to the expected active TE distribution across the entire genome (Fig. 3f). In detail, 93.8% of the DA LTRs were endogenous retroviruses (ERVs; Fig. 3g). In line with our findings so far, a motif analysis conducted on just the DA LTRs also revealed enrichment for the JUN binding motif (e-value = $1.3 \times 10^{-165}$, Fig. 3h).

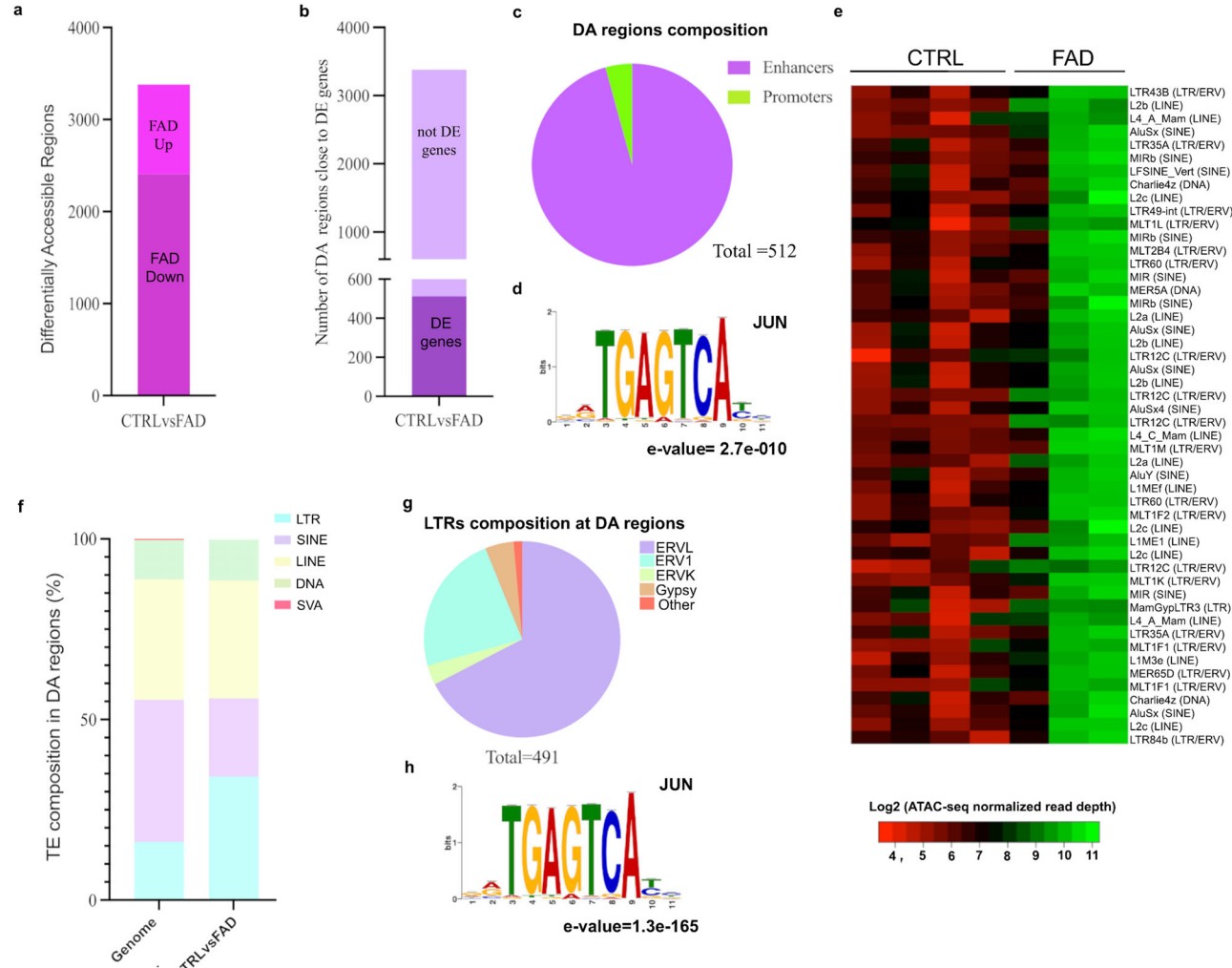

**Fig. 3 | Differentially accessible transposable elements in FAD hpNPCs.**
**a** Differentially accessible (DA) regions in the FAD hpNPCs compared to CTRL hpNPCs. FAD Up = significantly more accessible in FAD relative to CTRLs; FAD Down = significantly less accessible in FAD relative to CTRLs. **b** DA regions located near a differentially expressed gene. **c** DA regions near differentially expressed genes are predominantly enhancers (>1 kb from the transcription start site [TSS]). **d** MEME-ChIP analysis of 3382 DA regions uncovered the JUN binding motif as enriched. **e** Heatmap showing the top 50 TE copies identified as significantly more accessible in FAD than CTRL hpNPCs. One of the FAD samples was removed from the analysis and heatmap, as preliminary PCA revealed it behaved as an outlier, suggesting a technical issue with the sample. **f, g.** Family distribution of the aberrantly active TEs in FAD progenitors. **h** The aberrantly active LTRs are enriched for the JUN motif.

These data indicate that the FAD progenitors are characterized by dysregulated chromatin accessibility at thousands of genomic sites, including hundreds of TEs. Moreover, our data suggests that most of these genomic regions are c-Jun target sites and that aberrant c-Jun activity may underlie this observed chromatin dysregulation. For all these reasons, we further investigated the role of c-Jun in this aberrant pathway.

**Aberrant TE mobilization leads to the cytoplasmic accumulation of RNA–DNA hybrids in FAD hippocampal progenitors**
Several studies have demonstrated that aberrant TE de-repression and mobilization are observed across various neurodegenerative disorders, including AD[37–39,41–44]. A recent study in blind mole rats (an aging model for the study of longevity) has shown that TE de-repression leads to the cytosolic accumulation of TE-derived RNA–DNA hybrids, which activates the cGAS–STING innate immune signaling pathway leading to cell death[69]. Thus, we set out to investigate if the aberrant TE de-repression observed in the FAD hpNPCs also leads to the cytoplasmic accumulation of RNA–DNA hybrids, innate immune response, and cell death.

Intriguingly, immunostaining conducted on FAD and CTRL progenitors with the S9.6 antibody, specific for the detection of RNA–DNA hybrids[69], displayed significant hybrid accumulation in the cytoplasm of both FAD lines relative to the CTRLs (Fig. 4a, b). To confirm the specificity of the S9.6 antibody for RNA–DNA hybrids, we treated FAD hpNPCs with Ribonuclease H (RNase H), as this enzyme selectively degrades RNA–DNA hybrids. The treatment resulted in a significant decrease of these RNA–DNA hybrids relative to untreated progenitors of the same line (Supplementary Fig. 3c). Additionally, the RNA–DNA hybrid accumulation in FAD hpNPCs could also be facilitated by the downregulated endogenous RNase H expression levels observed in the FAD progenitors (Fig. 4c)[70].

Finally, to functionally validate which TEs are generating the RNA–DNA hybrids, we isolated the cytoplasmic fraction of the CTRL and FAD progenitors and performed DNA-qPCR using primers specific for distinct TE families. We identified a significant increase in cytoplasmic HERVK DNA in FAD relative to CTRL (Fig. 4d), suggesting that most of the detected RNA–DNA hybrids are derived from this ERV family. This finding is consistent with our previous genomic data that identified LTRs, specifically ERVs, in DA regions in FAD progenitors.

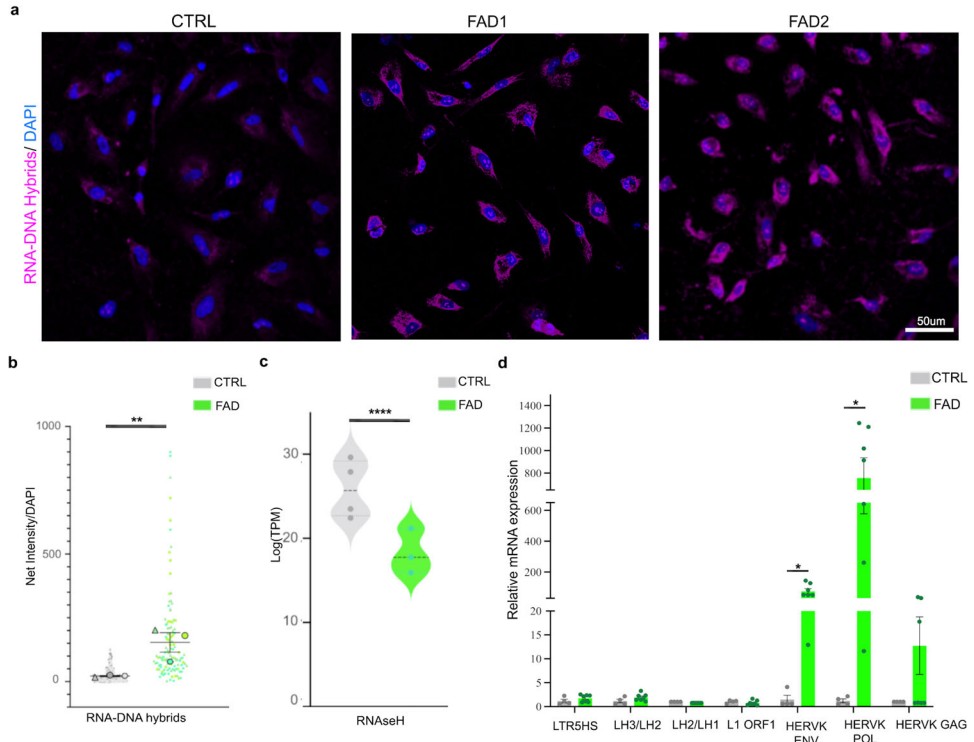

**Fig. 4 | Aberrant TE mobilization leads to the cytoplasmatic accumulation of RNA–DNA hybrids in FAD hippocampal progenitors. a** Immunofluorescence for RNA–DNA hybrids (S9.6 antibody, pink signal) displays an accumulation of RNA–DNA hybrids in the cytoplasm of FAD hpNPCs. Scale bar 50 μm, 40× magnification. DAPI staining on nuclei in blue. Each IF was replicated three times with similar results ($n = 3$ independent experiments). **b** Quantification of immunostaining in panel (**a**). The IF data are quantified using a Superplot, which concisely visualizes individual data points and their averages. The distinct combinations of colors and shapes indicate the three independent experiments performed. Each small dot in the graph corresponds to a specific data point representing an analyzed image or cells. The larger dots represent the average values calculated from the respective data points. A Repeated measures ANOVA test was used for the comparison between the two groups. A value of $P < 0.05$ was considered significant;

**$P < 0.01$ (RNA;DNA hybrids $P = 0.00329$). Error bars report the standard error. **c** Violin plot of $\log_2(\text{TPM})$ for RNaseH in CTRL and FAD hpNPCs. **d** qPCR analysis on the cytoplasmic fraction for a group of TEs selected among those previously identified as aberrantly active in FAD hpNPC. Primers for HERVK/LTR5HS target individual ORFs from the LTR; Primers for L1-ORF1 target a conserved region in ORF1 of 6x; the other L1 primers target the L1PA2 family; the LH2/LH3 primers target the end of the 3′ UTR; The LH1/LH2 primers target the 5′ of the other amplicon. Each dot in the bar plot represents an experiment ($n = 7$ independent experiments). A two-sided $t$-test was used for the comparison between the two groups. A value of $P < 0.05$ was considered significant; *$P < 0.05$ (HERVK ENV $P = 0.013637$; HERVK POL $P = 0.012468$); n.s., not significant (LTR5HS $P = 0.245766$; LH3/LH2 $P = 0.147473$; LH2/LH1 $P = 1$; L1 ORF1 $P = 0.366467$; HERVK GAG $P = 0.184794$). Error bars report the standard error.

Interestingly, recent studies have shown that HERVK is the only human ERV that retains a functional POL and thus is putatively still able to retrotranscribe cDNA[71–73]. Cumulatively, these experiments revealed that the FAD progenitors display aberrant cytoplasmic accumulation of HERVK-derived RNA–DNA hybrids.

## The accumulation of cytosolic RNA–DNA hybrids activates the cGAS–STING pathway

In mammalian cells, the presence of cytosolic DNA is sensed by the cGAS–STING pathway, which coordinates an immune response, resulting in the production of interferon-gamma (INF-γ)[74,75]. In addition to INF-γ response, the cGAS–STING pathway has recently been linked to cell senescence and cell death[76]. Consistent with this premise, we detected cGAS–STING pathway activation in FAD progenitors via immunostaining (Fig. 5a, b) and immunoblotting (Fig. 5c) through increased levels of STING and cGAS respectively. In line with the previous studies suggesting that cGAS–STING drives INF-γ production and cell death[77,78], we observed an increase in both INF-γ and cleaved caspase 3 (CC3) in the FAD progenitors (Fig. 5c, d). Additionally, the inhibition of STING in FAD progenitors (using the H151 compound[79]) led to a significant reduction of CC3 levels relative to the untreated FAD progenitors (Supplementary Fig. 3d). This decrease in CC3 levels upon STING inhibition in FAD demonstrates the causal link between cGAS–STING pathway activation and cell death.

These experiments provide a mechanistic link between aberrant TE mobilization, cytoplasmic accumulation of RNA–DNA hybrids, and cGAS–STING activation, resulting in inflammation and caspase-3 activation in FAD progenitors.

## Validation of the c-Jun-TE-cGAS–STING axis in a FAD isogenic system

We aimed to unequivocally demonstrate that the observed cellular and molecular phenotypes, including c-Jun upregulation, TE mobilization, RNA–DNA hybrid formation, and cGAS–STING pathway activation, were not an artifact of the genetic background of the different iPSC lines used for the experiments.

For this purpose, we differentiated an additional isogenic FAD iPSC pair (hereafter FAD3 and iso_CTRL3) into hpNPCs[45] using the same approach previously described (Fig. 1a). As expected, the FAD3 hpNPCs displayed impaired neurogenesis, as exhibited by a significant increase in TBR2- and FOXG1-positive intermediate progenitors as well as DCX-positive neuroblasts (Supplementary Fig. 4a). Additionally, FAD3 hpNPCs showed higher expression levels of c-Jun at both the gene (Supplementary Fig. 4b) and protein level (Supplementary Fig. 4c) relative to the isogenic control. Consistent with our previous results, we also observed higher expression levels of HERVK (Supplementary Fig. 4d) in the FAD3 progenitors. Moreover, immunostaining for RNA–DNA hybrids and STING demonstrated cytoplasmic

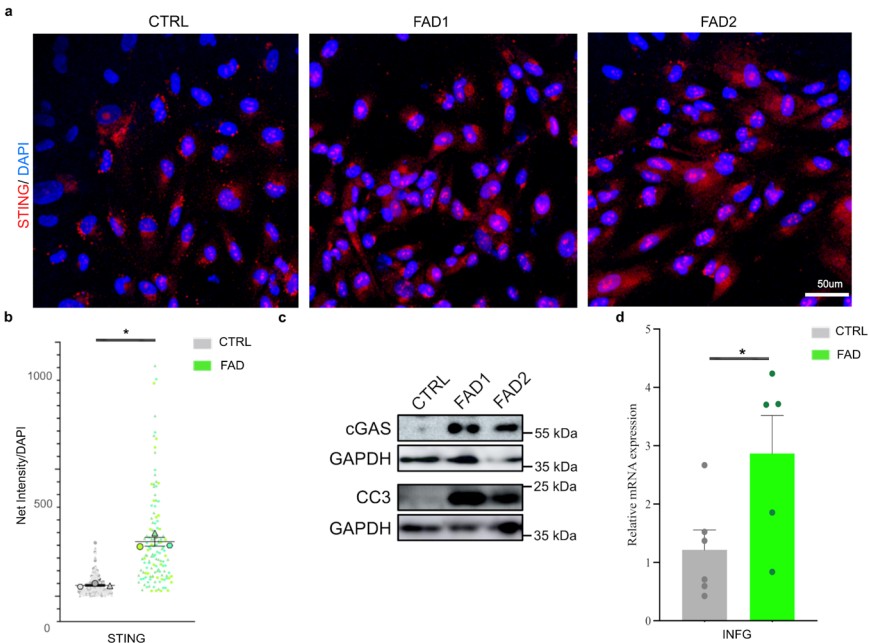

**Fig. 5 | Accumulation of RNA−DNA hybrids triggers the cGAS−STING cascade and CC3 activation in FAD hpNPCs. a** Immunofluorescence for STING (red signal) shows the and an upregulation of STING in FAD hpNPCs. Scale bar 50 μm, 40× magnification. DAPI staining on nuclei in blue. Each IF was replicated three times with similar results (*n* = 3 independent experiments). **b** Quantification of immunostaining in panel a. The IF data are quantified using a Superplot, which concisely visualizes individual data points and their averages. The distinct combinations of colors and shapes indicate the three independent experiments performed. Each small dot in the graph corresponds to a specific data point representing an analyzed image or cells. The larger dots represent the average values calculated from the respective data points. A repeated measures ANOVA test was used for the comparison between the two groups. A value of *P* < 0.05 was considered significant; *P* < 0.05 (STING *P* = 0.0104). Error bars report the standard error. **c** Immunoblots for cGAS and cleaved caspase 3 (CC3) in FAD and CTRL hpNPCs. The blot was replicated three times with similar results (*n* = 3 independent experiments). **d** Relative mRNA expression of interferon-gamma via qPCR. Each dot in the bar plot represents an experiment (*n* = 5 independent experiments). A two-sided *t*-test was used for the comparison between the two groups. A value of *P* < 0.05 was considered significant; *P* < 0.05 (INFG *P* = 0.0418). Error bars report the standard error.

RNA−DNA hybrid accumulation and upregulation of STING in the FAD3 progenitors relative to the isogenic control (Supplementary Fig. 4e).

Notably, both SOX2- or NESTIN-positive early FAD progenitors and DCX-positive neuroblasts express c-Jun, exhibited cytoplasmic RNA−DNA hybrid accumulation, and activate STING (Supplementary Fig. 5), suggesting that the observed phenotypes are not to be attributed to the varying genetic backgrounds of the FAD and CTRL progenitors.

In summary, the experiments conducted on the isogenic pair completely replicate and support all the main findings so far, suggesting that the herein-discovered c-Jun-TE-cGAS−STING axis is a mechanism of the disease.

### c-Jun inhibition reduces the neurogenic defects observed in the FAD hippocampal progenitors

Our genomic data revealed a distinct role for c-Jun in AD. Many of the differentially expressed genes are known c-Jun-regulated genes, and many DA regions harbor the JUN binding motif. Our data also showed an increase in *JUN* expression in the FAD progenitors, and the activation of the WNT/JNK pathway may trigger phosphorylation of the upregulated c-Jun, leading to increased aberrant c-Jun activation. We hypothesize that these processes may lead to aberrant AP-1 activity, resulting in the opening of thousands of genomic regions harboring the JUN binding motif, allowing for the de-repression of hundreds of TEs that are typically repressed in neural precursors.

To functionally validate our genomic data and test this hypothesis, we treated CTRL and FAD hpNPCs with a synthetic peptide competitor for binding JNKs, called c-Jun peptide (see "Methods"). This peptide disrupts the interaction between JNK and c-Jun, ultimately inhibiting c-Jun phosphorylation and, therefore pathway activation

(Fig. 6a). Notably, treatment of the progenitors with the c-Jun inhibitor for five days led to a partial reduction of the neurogenic defects previously observed (Fig. 1) in the FAD progenitors. Namely, in the c-Jun inhibitor-treated FAD progenitors (hereafter FAD+c-Jun peptide), the expression of *TBR2* and *FOXG1* was comparable to the CTRL (Fig. 6b), suggesting a recovery of the intermediate progenitor pool.

Next, we performed RNA-seq on FAD+c-Jun peptide and untreated FAD progenitors. This experiment led to the identification of 1106 differentially expressed genes (FDR < 5%, $\log_2$(FC) = ± 1.5; Fig. 6c). Notably, nearly a third of these genes (354/1106) were previously identified as differentially expressed when comparing CTRL and FAD progenitors (Fig. 6d), suggesting that inhibiting c-Jun phosphorylation significantly reduces the transcriptomic aberrations observed in the FAD hpNPCs.

Significantly, 63 genes previously downregulated in FAD were ultimately "rescued" upon the inhibition of c-Jun phosphorylation (Supplementary Table 2). Pathway analysis on these genes revealed enrichment for neuro-processes as well as inflammatory response and cell death (Fig. 6e). Using the Reactome database[72] to further analyze the identified pathways in detail, we identified enrichment for various specific inflammatory responses, including INF induction and activation, regulation of innate immune responses to cytosolic DNA, and STING-mediated immune response (Fig. 6f).

Altogether, these data support that c-Jun plays a crucial role in triggering inflammatory phenotypes characteristic of AD.

### c-Jun inhibition rescues TE de-repression, RNA−DNA hybrid formation, cGAS−STING activation, and caspase-3 activation

To investigate the role of c-Jun dysregulation in RTE aberrant re-activation, we performed RT-qPCR on a group of RTEs selected among those previously identified as aberrantly active in FAD. We observed a

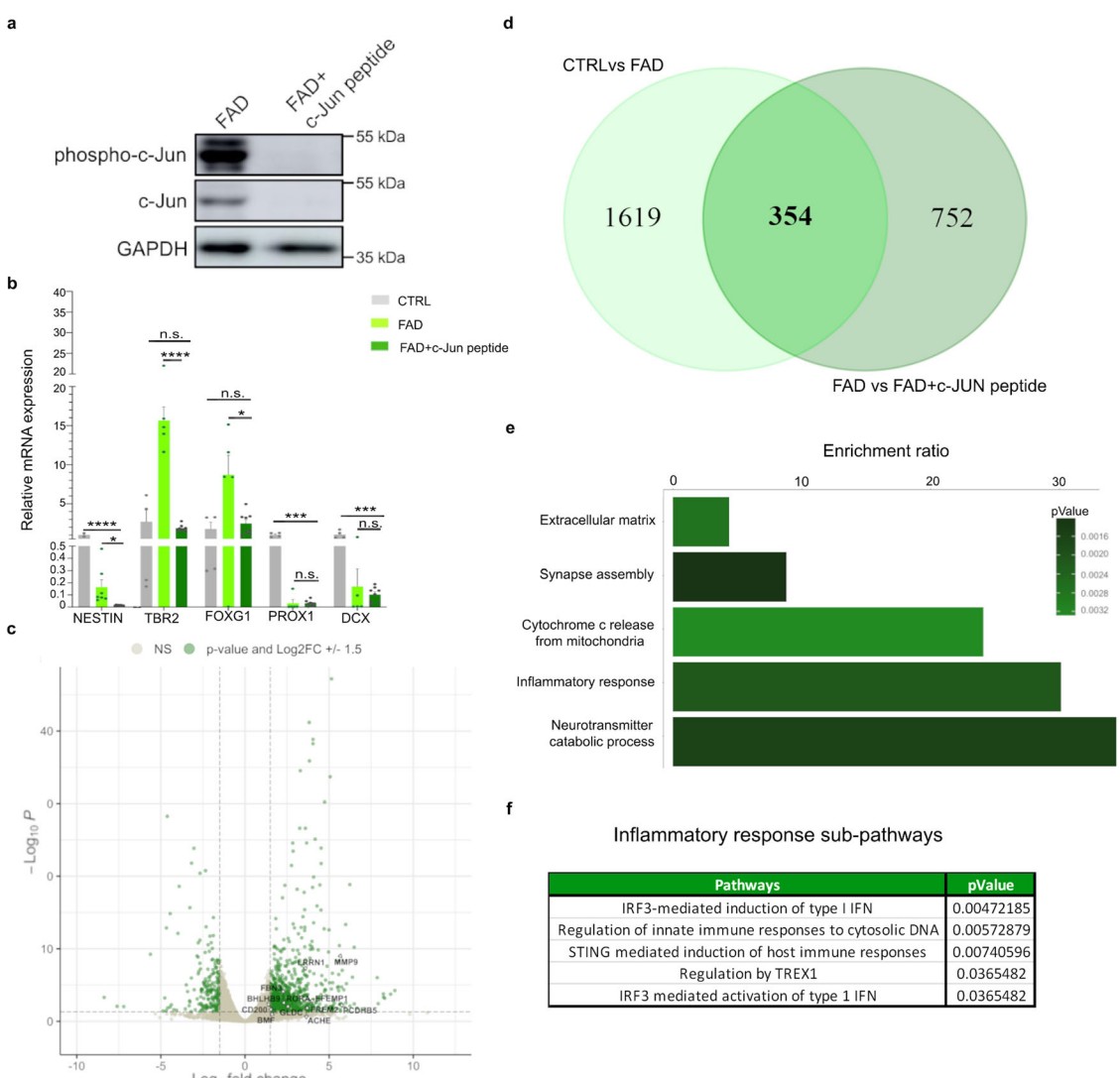

Fig. 6 | **Inhibiting c-Jun phosphorylation partially rescues the impaired neurogenesis and the gene expression differences in FAD hpNPCs. a** Immunoblot of c-Jun and phosphorylated c-Jun in untreated FAD hpNPCs and FAD hpNPCs treated with c-Jun peptide (FAD+c-Jun peptide). The blot was replicated three times with similar results (*n* = 3 independent experiments). **b** qPCR analysis for hpNPC markers in CTRL progenitors, untreated FAD progenitors, and c-Jun peptide-treated FAD progenitors (FAD+c-Jun peptide). Each dot in the bar plot represents an experiment (*n* = 5 independent experiments). A two-sided *t*-test was used for the comparison between two groups. A value of *P* < 0.05 was considered significant; \**P* < 0.05 (Nestin FADvsFAD+c-Jun peptide *P* = 0.016535; FOXG1 FADvsFAD+c-Jun peptide *P* = 0.010804); \*\*\**P* < 0.001 (DCX CTRLvsFAD+c-Jun peptide P = 0.000168); \*\*\*\* P < 0.0001 (TBR2 FADvsFAD+c-Jun peptide P < 0.000001; NESTIN CTRLvsFAD +c-Jun peptide *P* = 0.000002; PROX1 CTRLvsFAD+c-Jun peptide *P* < 0.000001); n.s., not significant (PROX1 FADvsFAD+c-Jun peptide *P* = 0.993389; DCX FADvsFAD +c-Jun peptide *P* = 0.570373; TBR2 CTRLvsFAD+c-Jun peptide *P* = 0.442672, FOXG1

CTRLvsFAD+c-Jun peptide *P* = 0.465767). Error bars report the Standard Error. **c.** Volcano plot of differentially expressed genes in FAD+c-Jun peptide relative to untreated FAD (**e**). Labeled genes are the genes involved in the pathway analysis in panel (**e**). Green = differentially expressed genes passing significance thresholds p-value < 0.05 and log₂(fold-change) ± 1.5; Gray = not significant. A 5% false discovery rate (FDR) was used to correct for multiple testing. **d** The Venn diagram displays the genes that were differentially expressed in the "FAD+c-Jun peptide vs untreated FAD" comparison and in the "FAD vs CTRL" comparison. In total, 354 differentially expressed genes overlap in the two comparisons. (Venn diagram was made with https://bioinformatics.psb.ugent.be) **e** Pathways enriched in the 63 "rescued genes" post-c-Jun peptide treatment predicted by WebGestalt. A 5% false discovery rate (FDR) was used to correct for multiple testing. **f** Reactome pathway analysis on the 63 "rescued genes" post-c-Jun peptide treatment. A 5% false discovery rate (FDR) was used to correct for multiple testing.

significant reduction of RTE expression in the FAD progenitors treated with the c-Jun peptide relative to the same untreated line (Fig. 7a).

Notably, the FAD+c-Jun peptide progenitors also showed a significant reduction of RNA–DNA hybrid accumulation, cGAS–STING cascade activation, and CC3 levels relative to untreated cells (Fig. 7b, c; Supplementary Fig. 6a, b).

These experiments demonstrate that c-Jun dysregulation underlies nearly all the pathological cellular and molecular phenotypes observed in the AD samples.

## The c-Jun-TE-cGAS/STING axis is conserved across different AD types
To test whether the mechanism that we characterized in FAD progenitors is conserved between the two AD types (familial and sporadic) we differentiated two sporadic Alzheimer's iPSC lines (SAD1 and SAD2) into hpNPCs, following the same protocol previously described (Supplementary Fig. 7a). Notably, an RT-qPCR for the same markers characterized for FAD (NESTIN, TBR2, FOXG1, PROX1, and DCX) also revealed impaired neurogenesis in SAD, with an accelerated

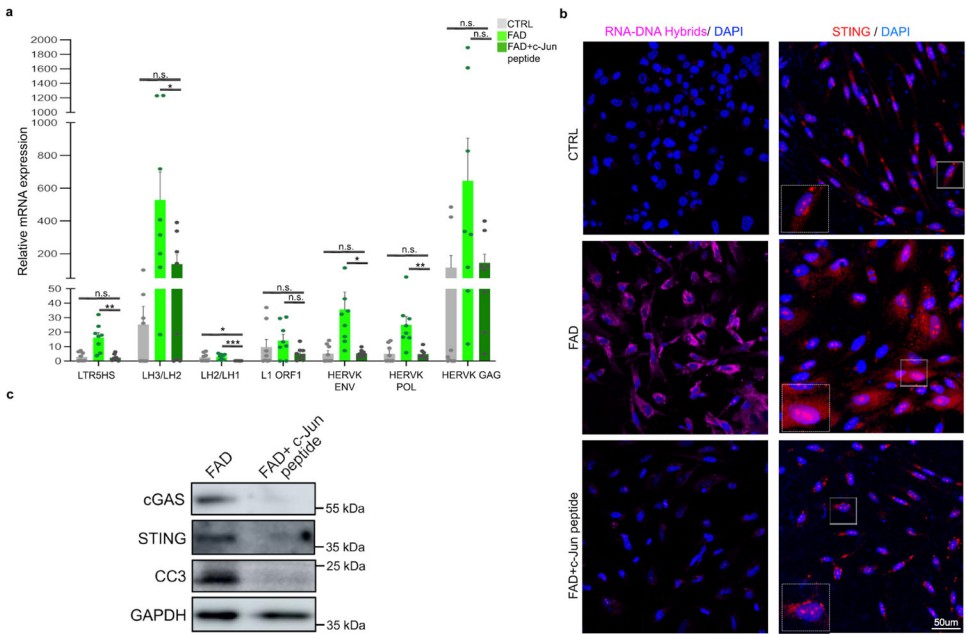

**Fig. 7 | Inhibiting c-Jun phosphorylation rescues aberrant TE derepression and the activation of the cGAS−STING cascade in FAD progenitors. a** qPCR analysis for a group of TEs selected among those previously identified as aberrantly active in FAD hpNPC. Primers for HERVK/LTR5HS target individual ORFs from the LTR; Primers for L1-ORF1 target a conserved region in ORF1 of 6×; the other L1 primers target the L1PA2 family; the LH2/LH3 primers target the end of the 3′ UTR; The LH1/LH2 primers target the 5′ of the other amplicon. Each dot in the bar plot represents an experiment (n = 8 independent experiments). A two-sided t-test was used for the comparison between two groups. A value of P < 0.05 was considered significant; *P < 0.05 (LH2/LH1 CTRLvsFAD+c-Jun peptide P = 0.028910; LH3/LH2 FADvsFAD+c-Jun peptide P = 0.046150; HERVK ENV FADvsFAD+c-Jun peptide P = 0.023226); **P < 0.01 (LTR5HS FADvsFAD+c-Jun peptide P = 0.001050; HERVK POL FADvsFAD+c-Jun peptide P = 0.003011) *** P < 0.001 (DCX CTRLvsFAD+c-Jun peptide P = 0.000168); ****P < 0.0001 (TBR2 FADvsFAD+c-Jun peptide P < 0.000001; NESTIN CTRLvsFAD+c-Jun peptide P = 0.000002; PROX1 CTRLvsFAD+c-Jun peptide P < 0.000001; LH2/LH1 FADvsFAD+c-Jun peptide P = 0.000514); n.s., not significant

(LTR5HS CTRLvsFAD+c-Jun peptide P = 0.733740; LH3/LH2 CTRLvsFAD+c-Jun peptide P = 0.073556, L1 ORF1 CTRLvsFAD+c-Jun peptide P = 0.409506; HERVK ENV CTRLvsFAD+c-Jun peptide P = 0.907864; HERVK POL CTRLvsFAD+c-Jun peptide P = 0.946895; HERV GAG CTRLvsFAD+c-Jun peptide P = 0.742228; HERV GAG FADvsFAD+c-Jun peptide P = 0.079540; L1 ORF1 FADvsFAD+c-Jun peptide P = 0.059455). Error bars report the standard error. **b** Immunofluorescence for RNA−DNA hybrids (S9.6 antibody, pink signal) and STING (red signal) shows that c-Jun inhibition significantly decreases the accumulation of RNA−DNA hybrids and STING levels in the cytoplasm of FAD hpNPCs. Scale bar 50 μm, 40× magnification. White dotted line boxes represent 2× magnification of the corresponding squared box. DAPI staining on nuclei in blue. Each IF was replicated three times with similar results (n = 3 independent experiments). **c** Immunoblots for cGAS, STING, and cleaved caspase 3 (CC3) on FAD+c-Jun peptide hpNPCs relative to untreated FAD progenitors. The blot was replicated three times with similar results (n = 3 independent experiments).

---

differentiation signature consistent with previous studies in sporadic AD[80] (Supplementary Fig. 7b). RNA-seq revealed that *JUN* is also upregulated in the SAD progenitors (Supplementary Fig. 7c). The 189 differentially expressed genes in SAD hpNPCs are involved in neurogenesis and neuronal differentiation pathways (Supplementary Fig. 7d). Moreover, as in FAD progenitors, SAD hpNPCs show abnormal chromatin accessibility at TE loci (Supplementary Fig. 7e).

Finally, treatment of SAD progenitors with the c-Jun inhibitor led to a decrease in cytoplasmic RNA−DNA dimers (Fig. 8a), as well as a significant reduction of cGAS−STING and cleaved caspase 3 activation (Fig. 8b).

These experiments complement those conducted in the FAD lines (including the isogenic pair) and unequivocally support that the pathological axis triggered by c-Jun dysregulation is a characteristic of AD, is conserved across AD types, and is not an artifact of the genetic and epigenetic heterogeneity of the different iPSC lines employed for this study.

## RTE-derived RNA−DNA hybrids and increased cGAS−STING activity are observed in familial and sporadic AD cerebral organoids

Finally, we tested whether this pathway was also active in cerebral organoids, which harbor progenitor cells and differentiated neurons in a three-dimensional architecture, modeling the human brain. The FAD, SAD, and CTRL iPSC lines were differentiated into cerebral organoids through an embryoid body intermediate. After 62 days, immunofluorescence was performed on cerebral organoids exhibiting the proper neuronal differentiation (Supplementary Fig. 8a). We observed a significant overall accumulation of RNA−DNA hybrids and an upregulation of STING in FAD and SAD organoids compared to CTRL organoids (Fig. 8c, Supplementary Fig. 8b). The activation of the cGAS−STING-cell death axis and the increase in CC3 levels were also confirmed through an immunoblot (Fig. 8d). As expected, the cytoplasmic accumulation of RNA−DNA hybrids was seen in TBR2-positive neural progenitors enriched in the FAD organoids (Supplementary Fig. 8c). Importantly, mature neurons (MAP2-positive) in the AD organoids also display a cytoplasmatic accumulation of RNA−DNA hybrids and an upregulation of STING (Supplementary Fig. 8d). Finally, activating the cGAS−STING cell-death axis in neurons leads to caspase-3 activation (Supplementary Fig. 8e). These organoid-based data further validated our proposed pathogenic mechanism and cascade for familial and sporadic AD. Moreover, these experiments revealed that the molecular impairment driven by TE re-activation observed in progenitors is also maintained in mature neurons in a physiological model of neural differentiation.

## Discussion

AD is the most common neurodegenerative disorder. Fibrillar deposits of highly phosphorylated TAU protein are a key pathological feature of AD and other AD-related dementias[81]. Importantly, studies in

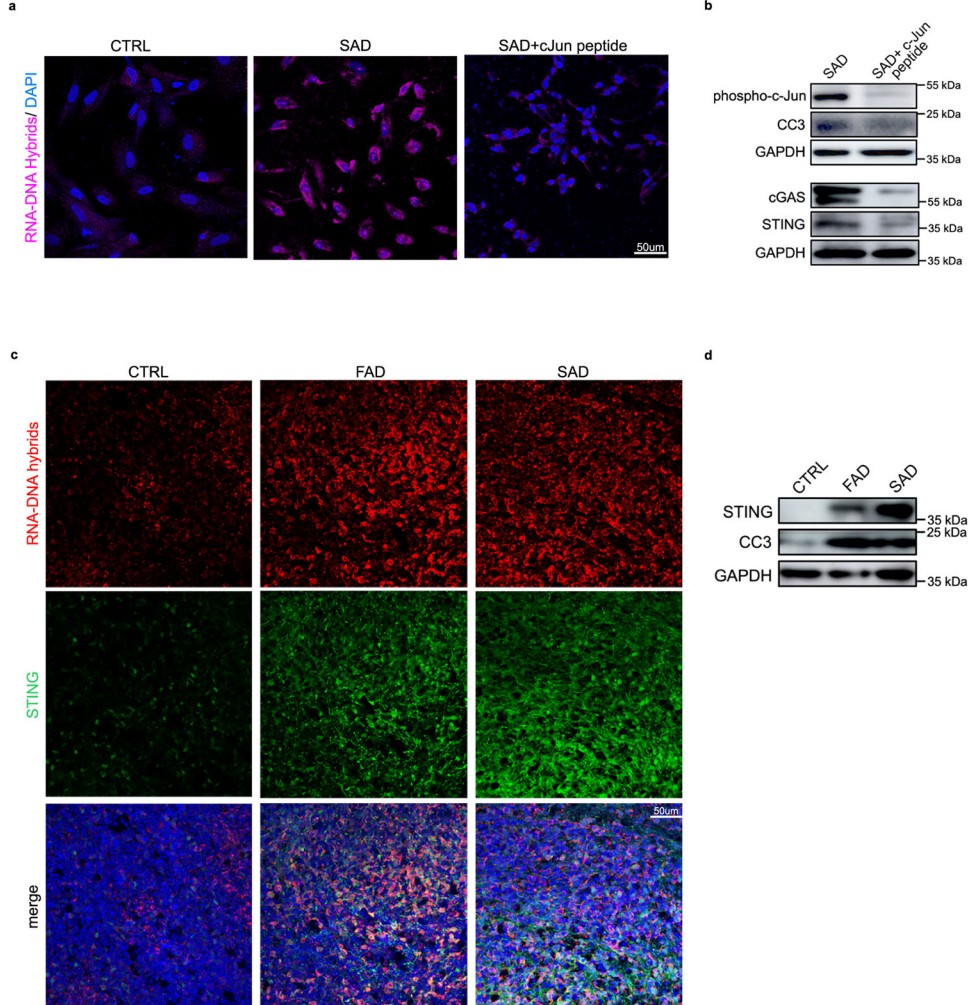

**Fig. 8 | TE-derived RNA–DNA hybrids and cGAS–STING activation in SAD progenitors and in AD cerebral organoids. a** Immunofluorescence for RNA–DNA hybrids (S9.6 antibody, pink signal) and STING (red signal) show that c-Jun inhibition significantly decreases the accumulation of RNA–DNA hybrids and STING levels in the cytoplasm of SAD hpNPCs. Scale bar 50 μm, 40× magnification. DAPI staining on nuclei in blue. Each IF was replicated three times with similar results (*n* = 3 independent experiments). **b** Immunoblots for phosphorylated c-Jun, cGAS, STING, and cleaved caspase 3 (CC3) on SAD+c-Jun peptide hpNPCs relative to SAD progenitors. The blot was replicated three times with similar results (*n* = 3 independent experiments). **c** Immunofluorescence for RNA–DNA hybrids (S9.6 antibody, red signal) and STING (green signal) on cerebral organoids (CTRL, FAD, SAD). Scale bar 50 μm, 40× magnification. DAPI staining on nuclei in blue. Each IF was replicated three times with similar results (*n* = 3 independent experiments). **d** Immunoblots for STING and cleaved caspase 3 (CC3) on AD and CTRL organoids. The blot was replicated three times with similar results (*n* = 3 independent experiments).

*Drosophila* have highlighted that Tau hyperphosphorylation and upregulation correlate with global nuclear chromatin relaxation and abnormal transcriptional activation of heterochromatic genomic regions[38,41]. One of the consequences of this phenomenon is the aberrant mobilization of TEs, which are typically repressed in the genome. Progressive TE de-repression and mobilization in the brain typically correlates with aging[37,42]. However, this phenomenon is significantly exacerbated in many neurodegenerative disorders, including amyotrophic lateral sclerosis (ALS), multiple sclerosis, and Alzheimer's[37,39–44].

Here, we unveiled a cascade of biological processes linking the upregulation of the AP-1 member, c-Jun, to aberrant TE mobilization. We detected an upregulation of c-Jun in hippocampal progenitors derived from familial and sporadic AD iPSC lines. Further, kinases involved in the regulation and phosphorylation of c-Jun (MAPK/JNK signaling) were also dysregulated in fully differentiated FAD iPSC-derived CA3 hippocampal neurons. We demonstrated that c-Jun upregulation has two main consequences: (1) the dysregulation of

hundreds of genes involved in neuronal differentiation and neuron generation and (2) the activation of hundreds of RTEs that harbor the AP-1 binding motif. Several recent studies have demonstrated that AP-1 can act as a pioneer transcription factor by recruiting the BAF chromatin remodeling complex to its targets to elicit accessibility and activation[82–84]. Thus, dysregulated AP-1 could trigger the opening of chromatin regions harboring its binding motif, potentially in cooperation with BAF. However, future experiments will be required to confirm this speculation. It is worth highlighting that the RNA-seq data generated in our study should be interpreted with caution, given that the impaired neuronal differentiation of the AD-derived iPSC lines may confound the gene expression comparison between healthy and AD samples due to potentially heterogeneous cell populations being compared. Nonetheless, we believe that studying the transcriptomic basis underlying the impaired neurogenesis is necessary to shed light on the molecular basis of the disease.

We then investigated the consequences of aberrant TE de-repression and mobilization. Our experiments demonstrated that

abnormal expression of RTEs leads to the accumulation of RTE-derived RNA–DNA hybrids in the cytoplasm of the AD hpNPCs and in AD cerebral organoids. This triggers the activation of the innate immune response, particularly of the cGAS–STING pathway, which ultimately elicits the accumulation of cleaved caspase-3, a molecular signature of cells undergoing apoptosis[85]. Importantly, we observed this phenomenon in both familial and sporadic AD lines. A recent study using Tau models revealed a pathogenic accumulation of ERV-derived RNA–RNA dimers in multiple brain cell types[86]. We were able to explicitly demonstrate that c-Jun facilitates this mechanism and that treating the AD hpNPCs with a c-Jun inhibitor sufficiently decreases this cascade, leading to a reduction of cell death.

Neuroinflammation plays an essential role in the pathogenesis of AD. In this context, the cGAS–STING signaling pathway has recently emerged as a critical mediator of inflammation in infection, cellular stress, and tissue damage[87]. Neuroinflammation is primarily driven by type-I interferons (INFs), and STING's role in controlling the type-I IFN-mediated response is becoming increasingly appreciated[88]. Given this premise, the findings of our studies may open new potential therapeutic avenues. For instance, nanobody-based targeting[89] of cytoplasmatic RNA–DNA hybrids might provide a new therapeutic approach to counteract the activation of innate immune response and to reduce neuroinflammation, bypassing the side effects of the anti-inflammatory drugs currently involved in the clinical trials. Additionally, experiments on pre-clinical models may be employed to test compounds that could act downstream in the cell death axis demonstrated here. Finally, the cytoplasmic accumulation of RNA–DNA hybrids could be used as an early biomarker for AD in diagnostic imaging tools.

In summary, these lines of evidence point toward a pathological mechanism underlying AD. Future studies on possible therapeutic interventions that would target this mechanism are essential to identifying therapeutic strategies and early diagnosis for AD and other neurogenerative disorders. It is worth mentioning that our RNA-seq data identified additional upstream regulators that are associated to the genes differentially expressed between Alzheimer's and control lines (e.g., WNT5A, MAPT, and TNF), but we only focused on c-Jun as its motif was the most enriched in the differentially accessible regions and in the differentially accessible TEs. Therefore, further studies on the other upstream regulators may reveal additional insights into the disease.

## Methods

### Human iPSC culture
Control and AD iPSC lines were obtained from the Coriell Institute for Medical Research (Camden, NJ). In particular, we received two control lines (Control line-1:IPSM8Sev3, male, 65 years old and Control line-2: iPSM15Sev4, female, 62 years old) CTRL1 and CTRL2 respectively; two familial AD lines (Familial Alzheimer's line-1: AG25370, female, 80 years old and Familiar Alzheimer's line-2: GM24675, male, 60 years old) FAD1 and FAD2 respectively; and two sporadic AD lines (Sporadic Alzheimer's line-1: AG27607, female, 69 years old and Sporadic Alzheimer's line-2: GM24666, male, 83 years old) SAD1 and SAD2 respectively. Finally, the isogenic iPSC pair FAD3/iso_CTRL3 was obtained by the Jackson Laboratories (revert mutant: JIPSC1054_PSEN2_N141I_REV/WT_human iPSC and Familiar AD mutant: IPSC, JIPSC1052_PSEN2_N141I_SNV/WT_human iPSC). The isogenic pair was generated from the same parental line KOLF2.1 J cell line (male). All the AD lines except for AG25370 were validated in previous studies[80,89–91].

The iPSC lines were expanded in feeder-free, serum-free mTeSR™1 medium (85850, STEMCELL Technologies). Cells were passaged ~1:10 at 80% confluency using EDTA 0.5 mM (15575020, Invitrogen), and small cell clusters (50–200 cells) were subsequently plated on tissue culture dishes coated overnight with Geltrex™ LDEV-Free hESC-qualified Reduced Growth Factor Basement Membrane Matrix (A1413302, Fisher-Scientific).

### hpNPC differentiation
The iPSC lines were differentiated into hpNPCs as previously described[45]. Briefly, iPSCs were treated with hpNPC induction medium for five days: DMEM/F-12 medium (Invitrogen) supplemented with B-27 (A3582801, Gibco), N-2 (17502048, Gibco), DKK1 (778606, Biolegend), Cyclopamine (C-8700, LC Laboratories), Noggin (597004, Biolegend), and SB431542 (S1067, Selleck Chemicals LLC). At day 6, the hpNPCs were plated in a new geltrex-coated well and cultured in proliferation medium, consisting of DMEM/F-12 medium (Invitrogen) supplemented with B-27 (A3582801, Gibco), N-2 (17502048, Gibco) and 20 ng/ml bFGF (713304, Biolegend).

### CA3 neuron differentiation
The iPSC lines were differentiated into CA3 Neurons as previously described[66]. Briefly, iPSCs were treated with hpNPC induction medium for 15 days. At day 16, the hpNPCs were plated in a new PLO-Laminin double-coated well in Neuron induction medium, consisting in DMEM/F12 medium (11320082, Gibco) supplemented with B-27 (A3582801, Gibco), N-2 (17502048, Gibco), BDNF (450-02, Prepotech), Dibutyryl-cAMP (11-415-0, Tocris), laminin (23017015, Thermofisher Scientific), AA (A4544-25G, Sigma), WNT3a (5036-WN, R&D System). After 3 weeks, the neurons were switched to neuron medium, consisting of DMEM/F-12 medium (Invitrogen) supplemented with B-27 (A3582801, Gibco), N-2 (17502048, Gibco), BDNF (450-02, Prepotech), Dibutyryl-cAMP (11-415-0, Tocris), laminin (23017015, Thermofisher Scientific) and AA (A4544-25G, Sigma) for one week. Mature CA3 neurons were then collected for RNA-seq and fixed for immunofluorescence.

### Cerebral organoid differentiation
CTRL and AD (FAD and SAD) iPSCs were differentiated into cerebral organoids following a previously published protocol[92]. Briefly, embryoid bodies were formed from CTRL, FAD, and SAD iPSCs and maintained in Essential 8 media (E8 media, A1517001, Thermo-scientific) supplemented with ROCK inhibitor (SCM075, Millipore) for 4 days. Neuronal induction was obtained by replacing the E8 media with Neural induction media: DMEM/F12 (11330-032, Invitrogen) supplemented with N-2 (17502048, Gibco), 1% Glutamax (35050-038, Invitrogen), 1% MEM-NEAA (M7145, Sigma), and Heparin at a final concentration of 1 µg/ml (H3149, Sigma). After 4–5 days, when neuroepithelium formation was achieved, spheroids were embedded in Matrigel (356234, BD Biosciences) and cultured in Cerebral organoid differentiation media: DMEM/F12 (11330-032, Invitrogen) and Neurobasal (21103049, Invitrogen) (1:1) supplemented with B27 without VitA (12587010, Invitrogen), N2 (17502048, Gibco), Insulin (I9278-5ML, Sigma), 2-Mercaptoethanol (1:100 dilution, 8057400005, Merk), 1% MEM-NEAA (M7145, Sigma), and Glutamax (35050-038, Invitrogen). After 4 days in static culture, spheroids were transferred to a shaker and maintained. Half media changes were performed every 3–4 days.

### hpNPC c-Jun peptide treatment
CTRL, FAD, and SAD hpNPCs were treated with 100 µM of c-Jun peptide (19-891, Fisher Scientific) for 5 days in a proliferative condition. This peptide comprises residues 33–57 of the JNK binding (δ) domain of human c-Jun and it is a competitive inhibitor of JNK/c-Jun interaction preventing c-Jun phosphorylation and activation.

### Treatment of hpNPCs with the H151 compound
FAD hpNPCs were treated with 1 µM of H151 compound (S6652, Selleckchem) for 5 days in a proliferative condition. H-151 is a highly potent and covalent antagonist of STING[79].

## Processing of organoids

At day 62, the whole organoids were fixed in 4% PFA overnight at 4 °C. After cryoprotection in 30% sucrose (s7903, Sigma), organoids were cryo-sectioned at 20 μm thickness and slices were analyzed by immunohistochemistry.

## Immunofluorescence

Immunohistochemistry of iPSCs, hpNPCs, and CA3 neurons was performed in μ-Slide 4 Well Glass Bottom (80426, IBIDI), while the IF of organoids was performed on 20-μm serial sections. Upon fixation (4% PFA for 10 minutes), cells were permeabilized in blocking solution (0.1% Triton X-100, 1× PBS, 5% normal donkey serum) and then incubated with the antibody of interest. The total number of cells in each field was determined by counterstaining cell nuclei with 4,6-diamidine-2-phenylindole dihydrochloride (DAPI; Sigma-Aldrich; 50 mg/ml in PBS for 15 min at RT). To improve the efficiency of TBR2 detection, the cells, and the organoid slides, before the permeabilization and blocking step, were treated with 10 mM sodium citrate (pH = 6) for 10 min at 95 °C.

For RNA–DNA hybrid staining (S9.6 antibody), upon fixation (4% PFA for 10 min), cells and organoid slides were permeabilized in PBS 1× 0.5% Triton X-100 for 15 min. They were then incubated overnight at −20 °C in 100% methanol. The samples were then blocked in 1× PBS 5% NDS for 4 h at 37 °C, followed by overnight incubation with the S9.6 antibody.

For the RNaseH experiment, upon fixation and permeabilization (as described above) the cells were treated with RNaseH (ab153634, Abcam, 1:50) overnight at 37 °C. The following day, an overnight incubation was performed at −20 °C in 100% methanol. The following day, the samples were blocked and incubated with the S9.6 antibody (as described above).

Immunostained cells and organoid slices were analyzed via confocal microscopy using a Nikon A1R+. Images were captured with ×40 for hpNPCs and ×20 and ×60 objectives for organoids and a pinhole of 1.0 Airy unit. Analyses were performed in sequential scanning mode to rule out cross-bleeding between channels. Fluorescence intensity quantification was performed with Fiji and the NIS-Elements AR software. In detail, we create a mask around each nucleus using DAPI intensity as the criterion. This mask served as the reference for measuring the fluorescence intensity of nuclear proteins. To analyze cytoplasmic proteins, the previously drawn nucleus mask was expanded to encompass the entire cytoplasm. All antibodies are listed in the Antibodies table (Supplementary Table S3).

## Western Blot

For total lysate, cells were harvested and washed three times in 1× PBS and lysed in RIPA buffer (50 mM Tris-HCl pH7.5, 150 mM NaCl, 1% Igepal, 0.5% sodium deoxycholate, 0.1% SDS, 500uM DTT) with protease and phosphatase inhibitors. Twenty μg of whole cell lysate were loaded in Novex WedgeWell 4–12% Tris-Glycine Gel (Invitrogen) and separated through gel electrophoresis (SDS-PAGE) in Tris-Glycine-SDS running buffer (Invitrogen). The proteins were then transferred to ImmunBlot PVDF membranes (ThermoFisher) for antibody probing. Membranes were incubated with 10% BSA in 1× TBST for 1 h at room temperature (RT), then incubated for variable times and concentrations with the suitable antibodies (Supplementary Table S3) diluted in 5% BSA in 1× TBST. Membranes were then washed with 1X TBST and incubated in the HRP-linked species-specific secondary antibody (1:10,000 dilution) for one hour at RT. The membrane was visualized using the Pierce ECL Plus Western Blotting Substrate (32132; ThermoFisher) and imaged with an Amersham Imager 680. All antibodies are listed in the Antibodies table (Supplementary Table S3).

## Isolation of cytoplasm

For cytosolic DNA isolation, a previously published protocol was followed (Shayla R. Mosley and Kristi Baker, 2022). Initially, the progenitors were harvested and washed twice in PBS. The cells were then resuspended in a buffer containing 10 mM Hepes pH 7.9, 25% glycerol, 1.5 mM MgCl$_2$, and 0.1 mM EDTA. After centrifugation at 6000$g$ for 10 minutes at 4 °C, the obtained cytosolic fraction (supernatant) was treated with Proteinase K (at a ratio of 0.55 mg of Proteinase K to 1 mg of total protein) to remove cytosolic proteins. Subsequently, a phenol-chloroform extraction was performed to eliminate Proteinase K. RNase A was used to remove RNA contaminants and another phenol-chloroform extraction was performed to eliminate RNase A. The resulting cytosolic DNA was utilized for qPCR.

## Real-time quantitative polymerase chain reaction (RT-qPCR)

Cells were lysed in Tri-reagent (R2050-1-50, Zymo Research) and RNA was extracted using the Direct-zol RNA Miniprep kit (Zymo Research). According to the manufacturer's directions, 600 ng of template RNA was retrotranscribed into cDNA using the RevertAid first-strand cDNA synthesis kit (Thermo Scientific). Totally, 15 ng of cDNA was used for each real-time quantitative PCR reaction with 0.1 μM of each forward and reverse primer, 10 μL of PowerUp™ SYBR™ Green Master Mix (Applied Biosystems) in a final volume of 20 μl, using a QuantStudio 3 Real-Time PCR System (Applied Biosystems). Thermal cycling parameters were set as follows: 3 min at 95 °C, followed by 40 cycles of 10 s at 95 °C and 20 s at 63 °C followed by 30 s at 72 °C. Each sample was run in triplicate. 18 S rRNA was used for normalization. Mitochondrial DNA was used for normalization in the qPCR in Fig. 4e. Primer sequences are reported in Supplementary Table S4.

## RNA-Seq sample preparation

Cells were lysed in Tri-reagent (R2050-1-50, Zymo Research) and total RNA was extracted using a Quick-RNA Miniprep kit (R1055, Zymo Research) according to the manufacturer's instructions. RNA was quantified using a DeNovix DS-11 Spectrophotometer while the RNA integrity number (RIN) was checked on an Agilent 2200 TapeStation. Only samples with RIN values above 8.0 were used for transcriptome analysis. RNA libraries were prepared using NEBNext® Poly(A) mRNA Magnetic Isolation Module (E7490S, New England Biolabs), NEBNext® UltraTM II Directional RNA Library Prep Kit for Illumina® (E7760S, New England Biolabs) and NEBNext® UltraTM II DNA Library Prep Kit for Illumina® (E7645S, New England Biolabs) according to the manufacturer's instructions. The libraries were sequenced using an Illumina NextSeq2000, generating 150 bp Paired-End reads.

## RNA-Seq analyses

Reads were aligned to hg19 using STAR v2.5[93] in two-pass mode with the following parameters: --quantMode TranscriptomeSAM --outFilterMultimapNmax 10 - -outFilterMismatchNmax 10 --outFilterMismatchNoverLmax 0.3 --alignIntronMin 21 -- alignIntronMax 0 --alignMatesGapMax 0 --alignSJoverhangMin 5 --runThreadN 12 -- twopassMode Basic --twopass1readsN 60000000 --sjdbOverhang 100. We filtered bam files based on alignment quality (q = 10) using Samtools v0.1.19[94]. We used the latest annotations obtained from Ensembl to build reference indexes for the STAR alignment. Kallisto[95] was used to count reads mapping to each gene. RSEM[96] was used to obtain FPKM (Fragments Per Kilobase of exon per Million fragments mapped). Differential gene expression levels were analyzed using DESeq2[97], with the following model: design = -batch + condition, where condition indicates either CTRL or AD (FAD or SAD) lines and batch indicates the technical replicate.

## ATAC-Seq sample preparation

For ATAC-Seq experiments, 50,000 cells per condition were processed as described in the original ATAC-seq protocol paper[98]. Briefly,

50,000 cells were collected, washed, and lysed. The chromatin was subjected to transposition/library preparation via a Tn5 transposase using the Tagment DNA Enzyme and Buffer Kit (20034197, Ilumina) and incubated at 37 °C for 30 min with slight rotation. Transposed DNA was purified using a MinElute PCR Purification Kit (28004; Qiagen). Transposed DNA fragments were then amplified using a universal and barcoded primer[98]. Thermal cycling parameters were set as follows: 1 cycle of 72 °C for 5 min, 98 °C for 30 s, followed by 5 cycles of 98 °C for 10 s, 63 °C for 30 s, and 72 °C for 1 min. The amplification was paused and 5 μl of the partially amplified, transposed DNA was used for a qPCR side reaction including the universal and sample-specific barcoded primers[98], PowerUp™ SYBR™ Green Master Mix (Applied Biosystems), NEBNext High-Fidelity 2× PCR Master Mix, and nuclease-free water. The qPCR side reaction parameters were set as follows: 1 cycle of 72 °C for 5 min, 98 °C for 30 s, followed by 40 cycles of 98 °C for 10 s, 63 °C for 30 s, and 72 °C for 1 min. The Rn vs cycle plot was used to determine the remaining number of PCR cycles needed where 1/3 of the maximum fluorescent intensity corresponds to the cycle number. The remaining partially amplified transposed DNA was fully amplified using the previous parameters with the additional cycle number determined from the qPCR side reaction. The amplified, transposed DNA was purified using AMPure XP beads (A63881, Beckman Coulter) and sequenced using an Illumina NextSeq2000, generating 150 bp Paired-End reads.

### ATAC-Seq analyses

After removing the adapters, the sequences were aligned to the reference hg19, using the Burrows–Wheeler Alignment tool (BWA), with the MEM algorithm[94]. Aligned reads were filtered based on mapping quality (MAPQ > 10) to restrict our analysis to higher quality and likely uniquely mapped reads, and PCR duplicates were removed. All mapped reads were offset by +4 bp for the forward strand and −5 bp for the reverse strand. We called peaks using MACS2[99], at 5% FDR, with default parameters. We analyzed differential genome accessibility using DESeq2[97], with the following model: design = ~batch + condition, where condition indicates CTRL or AD (FAD or SAD) lines and batch indicates the technical replicate. R v3.3.1. and BEDtools v2.27.1[100] were used for all comparative TEs analyses.

### Statistical and genomic analyses

All statistical analyses were performed using R v3.3.1. BEDtools v2.27.1[100] was used for genomic studies. Pathway analysis was performed with WEB-based GEne SeT AnaLysis Toolkit (http://www.webgestalt.org) and Reactome[101]. Motif analyses were performed using the MEME-Suite[102], specifically with the MEME-ChIP application. Fasta files of the regions of interest were produced using BEDTools v2.27.1. Shuffled input sequences were used as background. E-values < 0.001 were used as the threshold for significance. All described results (qPCR analyses and immunofluorescence) are representative of at least three independent experiments unless specifically stated otherwise. Data were presented as average ± SEM. Statistical analysis was performed using Excel (Microsoft) or GraphPad Prism 8 software (GraphPad). A repeated measures ANOVA test was used for the comparison between the two groups. A value of $P < 0.05$ was considered significant; *$P < 0.05$; **$P < 0.01$; ***$P < 0.001$; n.s., not significant.

The IF data are presented using a Superplot, which concisely visualizes individual data points and their averages. The distinct combinations of colors and shapes indicate the three independent experiments performed. Each small dot in the graph corresponds to a specific data point representing an analyzed image or cells. The larger dots represent the average values calculated from the respective data points.

### Reporting summary

Further information on research design is available in the Nature Portfolio Reporting Summary linked to this article.

## Data availability

The original genome-wide data generated in this study have been deposited in the GEO database under accession code GSE213610. Source data are provided with this paper.

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

## Acknowledgements

The authors are thankful to Julius Judd and Cedric Feschotte (Cornell University) for sharing the primers for RT-qPCR at TE loci and for insightful discussions of the findings. The authors thank the Jefferson Stem Cell Center for the preliminary expansion and characterization of the patient lines and for helping optimize the differentiation protocols. The authors thank Zoe Mitchell and Sophie Smith (Imperial College London) for reading and commenting an early version of the manuscript. This study was funded by NIH R35 GM138344-01 and Alzheimer's Association AARG-NTF-22-970553 grants, both awarded to M.T. The following sources also supported this work: NIH R21-NS090912 (D.T.), RF1-AG057882 (D.T.), Muscular Dystrophy Association grant 628389 (D.T.); DoD grant W81XWH-21-1-0134 (M.E.C.); and Farber Family Foundation (D.T. and C.S.).

## Author contributions

M.T., D.T., and C.S. designed the experiments. C.S. performed most of the experiments. S.M.B. and M.E.C performed some of the experiments. M.S. contributed to some experiments and analyses. C.S., M.T., S.M.B., and D.T. analyzed the data. C.S. wrote the paper with the contribution of all the authors.

## Competing interests

The authors declare no competing interests.
