## [Peer Review File · Nature Communications]

JUN upregulation drives aberrant transposable element mobilization, associated innate immune response, and impaired neurogenesis in Alzheimer's diseaseREVIEWER COMMENTS

Reviewer #1 (Remarks to the Author):

Adult neurogenic decline, inflammation, and neurodegeneration are the hallmarks of Alzheimer's disease (AD). However, less is understood about the mechanism for orchestrating these hallmarks. In this manuscript, Scopa et al. show that JUN drives transposable element mobilization, leading to the cytoplasmic accumulation of RNA- DNA hybrids and activation of the cGAS-STING pathway. And the activation of the cGAS-STING pathway is supposed to elicit the cell death. This is a pretty interesting manuscript. But, there are a few issues that need addressing:

Major Concerns

1. Scopa et al. show that cGAS and STING proteins are increased by JUN. But more work is still needed. The cGAMP or interferon levels should be assessed to determine whether the cGAS-STING pathway is activated.
2. The cGAS-STING pathway is potent to the stimulation of immune surveillance, but this pathway is less effective to promote cell death. I think that at this stage, the authors have not presented conclusive data. My suggestion would be to at least apply the cGAS or STING inhibitor to evaluate the contribution of the cGAS-STING pathway to the JUN-promoted cell death.
3. Sting is generally supposed to be located at cytoplasm (ER) as shown in Figure 7c. But, in Extended Data Figure 5d, STING is located dominantly in the nuclei. How then can the nuclear location of STING be explained?

Minor Concerns

1. Language need extensive editing. It is a bit awkward at places and will require extensive editing service.
2. The images of western blot and IF need be improved.

Reviewer #2 (Remarks to the Author):

The characterization of the role transposable elements, especially LINE1 and endogenous retroviruses, play during the reprogramming process, aging process, and pathology processes continues to emerge as a potentially crucial target for the benefit of human health and longevity. The results presented here by Scopa et. al show a correlation between an early Alzheimer's cellular pathology and transposon derepression, followed by the subsequent increase in cGas/STING protein abundance. Although these results appear to be quite promising, further experiments are needed to verify the claims of this manuscript, specifically that JUN activity drives TE mobilization and that this mobilization is causing inflammation via the cGas/STING pathway. In line 239 of the manuscript, the authors state that the S9.6 antibody is "specific for the detection of RNA-DNA hybrids." However, it has been shown that this antibody also binds nonspecifically to double-stranded RNA and can cause significant background during imaging.¹ In order to definitively state that the immunofluorescence in figure 4 is the result of hybrids formed via transposon activity, the authors must include a negative control treating control and FAD cells with RNase H prior to antibody treatment. Because RNase H specifically cleaves the RNA in RNA-DNA hybrids, this treatment can demonstrate the phenotype observed in FAD cells is not the result of double stranded RNA. Additionally, it would also serve the authors well to show that this signal is in fact the result of transposon expression. By fractionating these cells and performing DNA-qPCR on the cytoplasmic fraction between control and FAD cells, the authors can demonstrate a clear increase in transposon presence and can also differentiate between LINE1 and ERV composition. Finally, on line 245, the authors state cGas and STING are elevated in FAD cells via western blot, but the STING blot appears to be missing from the figure. Regarding the claim of inflammation via the cGas/STING pathway, additional experiments are needed to show that the increase in cGas/STING protein expression is directly resulting in

increased inflammation via this pathway. This can simply be shown via qPCR demonstrating elevated IFN expression in AD cells compared to controls. This will show that these cells are signaling for an inflammatory response. The authors could also utilize a commercially available cGAMP ELISA kit to show that cGas is binding to the hybrids and signaling towards STING activation.

This manuscript provides strong evidence towards the correlation between early Alzheimer's disease and JUN upregulation and transposon activation. The addition of the recommended controls will solidify the transposon aspects of this research.

1. Smolka, J.A., et al., Recognition of RNA by the S9.6 antibody creates pervasive artifacts when imaging RNA:DNA hybrids. *J Cell Biol*, 2021

Reviewer #3 (Remarks to the Author):

Scopa et al propose an interesting hypothesis for the role of aberrant c-JUN expression in early AD pathogenesis, which impacts transposable element activation, neurogenesis and the immune response. To investigate this mechanism, they primarily use iPSC-derived hippocampal progenitor cells, and conduct a few additional studies in iPSC-derived neurons and organoids.

While the premise is interesting and I appreciate the amount of work presented here, there are several major issues that give me cause for concern;

1. The majority of the manuscript is centered on 2xFAD vs 2xCTRL iPSC lines. Without an isogenic pairing or the use of multiple clones, this very small N means that it is very difficult to determine whether any difference between groups is actually due to FAD or due to the individual characteristics of each iPSC line (which are almost always of greater effect size than even an autosomal dominant mutation). The authors have identified many strikingly significant differences between groups throughout this work, however it does not appear that the correct statistical tests have been applied. The technical N has not been provided for any of the data presented - as an example Figure 1D, what is each data point? One well? One differentiation? What is each large dot? Presumably an average of each differentiation? I could not find this information for any figure throughout the whole manuscript. This brings me to the question about statistical tests - the methods describe an N=3 and a student's t-test used for analyses. This is not appropriate for the data - the authors have here an N=2 with multiple technical replicates. The stats as described are hugely inflating the significance of any test by essentially replicating data points and not taking into account that they are not independent observations. A more appropriate test to use while maintaining some power and variability would be a linear mixed model. Similarly, I could not find the N you used for your RNA-seq anywhere, but an equivalent approach, e.g. DREAM would be required to take account of technical replicates (which differentiations are) to avoid inflating effect sizes.

2. My second major concern is also related to the small N=2, with regards to their differentiation propensity. In the first figure the authors describe how CTRL and FAD lines differentiate into hpNPCs differently, with FAD lines seemingly generating more intermediate progenitors and fewer early neural stem cells. I do not think the authors have sufficient evidence to demonstrate that these differences are genuinely due to some pathogenic mechanism in FAD lines, rather than technical variation and differentiation capacity between different iPSC lines. iPSC lines and clones are notorious for highly variable propensities for differentiation into different cell types, and resulting cells have very variable levels of marker gene expression. Based on the data presented, the lack of isogenic controls and the small N, I don't think we can be confident that this is specific to FAD lines. More work needs to be done to dig into this further - e.g. in the absence of isogenic/more lines: utilize the RNA-seq data to dig more into cellular identity/PCA, try different differentiation protocols, direct neuronal induction, sequence the iPSCs etc

3. Unfortunately, the concerns in point no.2 colour interpretation of the rest of the manuscript. If the cellular populations are so different between CTRL and FAD, how can you justify comparing the

two on a bulk level by RNA-seq? How many gene expression changes are due to FAD and how many are due to different cell types? e.g. WNT and JUN expression are essential for neuronal differentiation - do you see higher expression of JUN because the FAD lines are better at differentiating and are more mature than the CTRLs? Similarly for differentiation pathways found in the RNA-seq - is it a disease-associated mechanism or reflective of different cellular populations being sequenced? This extends into the neuronal and organoid data - it is not surprising that neuronal populations would differ given that the NPCs are not the same. A generic neuronal marker would be helpful for Fig.2 to determine if you are seeing fewer neurons generally (and more glia) or specifically fewer CA3 neurons in the FAD lines. Was the subsequent RNAseq corrected for different CA3 proportion? How much is disease phenotype and how much is stemming from iPSC line differentiation capacity? The authors point out MAPT as a gene of interest spanning all pathways, however MAPT is also a useful generic neuronal marker, and again could be popping up due to differences in cell type proportions between cultures, rather than being differentially expressed in FAD lines. Differences in differentiation capacity and maturity may also underlie the ATAC-seq and TE data - it has been reported that there is TE de-repression during neuronal differentiation and substantial remodeling of chromatin - are the differences in these things due to an AD pathogenic mechanism, or technical artefact in differentiation capacity?

Overall, the manuscript and hypothesis are very interesting, and may well be correct. However, given the experimental design and statistical analyses used, at this point the conclusions are difficult to justify given the nature of the iPSC line characteristics, differentiation differences and N.

Other comments:

1. There are many places throughout the manuscript where there is some over-interpretation in the text, e.g. where is the evidence for AP-1 and BAF recruitment in these data? Where is demonstration that activating cGAS-STING induces caspase 3 activation? These things may be associated in the literature, but be careful to not state you have demonstrated them in these data.
2. Validation of your apoptosis hypothesis leading from caspase activation should be straightforward to do with IF. Do you think the different cell populations resulting from differentiation are due to increased cell death or altered differentiation? Or both?
3. Given that you identify STING and its involvement in interferon responses, is it worth going back to your very first IPA analysis where interferon signaling was the most significant pathway? What's the relevance of it?
4. Around figure 4, you stop presenting both CTRL lines - are both still being used? Why not include that data?
5. It would be useful to co-stain STING/RNA-DNA hybrid images with specific population markers, to see if they are specific to more mature progenitor populations
6. Figure 5, referring to these data as "rescue" of the IP pool feels misleading, as you are reducing their number (or at least expression of the markers) rather than rescuing them. There is also even lower Nestin expression and little change in your other markers - so what is actually happening? What are the cells if not increasing the stem cell progenitor pool? Related to major concerns - is it really surprising that inhibiting c-JUN, a regulator of differentiation, reduces differentiation?
7. Make sure to check references are correct - 86 in the methods should be 87. I did not go through and check others. How do Lancaster organoids relate to hpNPCs and hippocampus? How did you determine what "proper" neuronal differentiation was? The FADs and SADs actually have much nicer looking rosettes than controls, and may be better at making neurons (consistent with all your other data). The IF data here is also odd (particularly in controls) - I wouldn't expect SOX2 to have any overlap with NeuN and CTIP2.
8. Overall, what is your proposed mechanism for upregulated JUN and dysregulated chromatin accessibility in FAD and SAD lines? They presumably have no pathology yet, MAPT expression is low, is it Ab processing? Is it a stress response - to what?

ASNWERS TO REVIEWER COMMENTS

Reviewer #1 (Remarks to the Author):

Adult neurogenic decline, inflammation, and neurodegeneration are the hallmarks of Alzheimer's disease (AD). However, less is understood about the mechanism for orchestrating these hallmarks. In this manuscript, Scopa et al. show that JUN drives transposable element mobilization, leading to the cytoplasmic accumulation of RNA- DNA hybrids and activation of the cGAS-STING pathway. And the activation of the cGAS-STING pathway is supposed to elicit the cell death. This is a pretty interesting manuscript. But, there are a few issues that need addressing:

We sincerely appreciate Reviewer-1's valuable feedback and positive assessment of our manuscript. In response to Reviewer-1's suggestions, we have revised the manuscript, addressing each of the highlighted concerns in a comprehensive manner. Below, we outline the specific changes made:

Major Concerns

1. Scopa et al. show that cGAS and STING proteins are increased by JUN. But more work is still needed. The cGAMP or interferon levels should be assessed to determine whether the cGAS-STING pathway is activated.

Thank you for your comment. We performed qPCR analysis to evaluate interferon gamma levels, as suggested. The results of this analysis corroborate our model and align with the RNA-seq data presented in Figure 2b. Specifically, we observed a significant increase in interferon levels in FAD samples in comparison to controls. We are confident that this analysis of interferon levels further strengthens our findings surrounding JUN and cGAS-STING-cell death axis activation.

These findings have been integrated into the revised manuscript, particularly in the revised Figure 5e and reported here for the Reviewer's convenience.

2. The cGAS-STING pathway is potent to the stimulation of immune surveillance, but this pathway is less effective to promote cell death. I think that at this stage, the authors have not presented conclusive data. My suggestion would be to at least apply the cGAS or STING inhibitor to evaluate the contribution of the cGAS-STING pathway to the JUN-promoted cell death.

We agree with Reviewer-1 that the cGAS-STING pathway is primarily involved in stimulating innate immunity through the regulation of IFN production. However, emerging evidence has also highlighted the role of the cGAS-STING pathway in promoting apoptosis upon activation [PMID: 32990676; PMID: 36769349].

As the reviewer suggested, we treated FAD progenitors with a well-established STING inhibitor: H151 compound [PMID: 29973723]. Notably, STING inhibition led to a significant reduction of cleaved caspase-3 levels. We believe that these findings provide

clear evidence that the cGAS-STING pathway promotes apoptosis in FAD progenitors, and we have incorporated these data into the revised manuscript in Extended Data Figure 3d. We also reported the data here for the convenience of the reviewer.

3. Sting is generally supposed to be located at cytoplasm (ER) as shown in Figure 7c. But, in Extended Data Figure 5d, STING is located dominantly in the nuclei. How then can the nuclear location of STING be explained?

Thank you for raising an important point regarding the subcellular localization of STING.

In response to the Reviewer's observation, we agree that the prevalent consensus is that STING is primarily localized in the cytoplasm as demonstrated in Figure 7c. However, we acknowledge that the Extended Data Figure 5d shows a predominant nuclear localization of STING. We'd like to highlight that multiple recent papers have reported evidence of nuclear localization for both cGAS [PMID: 30846571; PMID: 30811988] and STING [PMID: 27791205; PMID: 34541469; PMID: 34625708] that correlates with their immune surveillance function. Additionally, other studies have identified nuclear STING localization in various cell types by immunofluorescent staining, including cancer cells [PMID: 27791205; PMID: 36109513] and neurons within the context of neurodegenerative diseases [PMID: 32253733; PMID: 37239045]. As seen in our manuscript and others, differences in the localization of STING does not affect its role in immune surveillance, and ultimately cell death. However, uncovering mechanisms surrounding STING subcellular localization is outside the scope of this paper.

Minor Concerns

1. Language need extensive editing. It is a bit awkward at places and will require extensive editing service.

We would like to emphasize that two of the authors are native English speakers. Nonetheless, in an effort to improve the linguistic quality of the manuscript, the revised version of the manuscript underwent thorough editing by two additional native English speakers (see acknowledgements).

2. The images of western blot and IF need be improved.

We have enhanced the quality of all the Western blot and IF images presented throughout the manuscript.

Reviewer #2 (Remarks to the Author):

The characterization of the role transposable elements, especially LINE1 and endogenous retroviruses, play during the reprogramming process, aging process, and pathology processes continues to emerge as a potentially crucial target for the benefit of human health and longevity. The results presented here by Scopa et. al show a correlation between an early Alzheimer's cellular pathology and transposon derepression, followed by the subsequent increase in cGas/STING protein abundance. Although these results appear to

be quite promising, further experiments are needed to verify the claims of this manuscript, specifically that JUN activity drives TE mobilization and that this mobilization is causing inflammation via the cGas/STING pathway.

We are pleased to note Reviewer 2's appreciation of the manuscript. In response to their comments, we have addressed all the concerns highlighted. We believe that these revisions have significantly improved our findings.

To provide more context, we have detailed the specific changes made in response to the Reviewer's comments below:

1) In line 239 of the manuscript, the authors state that the S9.6 antibody is "specific for the detection of RNA-DNA hybrids." However, it has been shown that this antibody also binds nonspecifically to double-stranded RNA and can cause significant background during imaging. In order to definitively state that the immunofluorescence in figure 4 is the result of hybrids formed via transposon activity, the authors must include a negative control treating control and FAD cells with RNase H prior to antibody treatment. Because RNase H specifically cleaves the RNA in RNA-DNA hybrids, this treatment can demonstrate the phenotype observed in FAD cells is not the result of double stranded RNA.

We agree with the reviewer that the specificity of the S9.6 antibody for detecting RNA-DNA hybrids is still under debate. Considering that the potential for nonspecific binding to double-stranded RNA is an important point in our study, we addressed this concern by treating our FAD cells with RNase H. We are pleased to report that this treatment resulted in a significant decrease in the RNA-DNA hybrid signal detected by the S9.6 antibody (p -value= 0.0216). This outcome supports the specificity of the antibody for RNA-DNA hybrids in our system and validates our previous findings.

The observed immunofluorescence signal in Figure 4 is indeed the result of RNA-DNA hybrids and not an artifact caused by double-stranded RNA binding.

We have included the results of this control experiment in the new Extended Data Figure 3. We also report the data here for Reviewer's convenience:

2) Additionally, it would also serve the authors well to show that this signal is in fact the result of transposon expression. By fractionating these cells and performing DNA-qPCR on the cytoplasmic fraction between control and FAD cells, the authors can demonstrate a clear increase in transposon presence and can also differentiate between LINE1 and ERV composition.

We thank Reviewer 2 for highlighting the importance of demonstrating that the observed RNA-DNA hybrids signal indeed originates from transposon expression.

We followed Reviewer 2's recommendation and performed DNA-qPCR analyses comparing the cytoplasmic fractions of control and FAD cells.

This experiment yielded significant results consistent with specific TE mobilization and cytoplasmic accumulation. Specifically, the DNA-qPCR analysis revealed a clear and

significant increase in HERVKs in FAD samples compared to control. HERVK is the only human ERV to retain functional RT capabilities [PMID: 34166614; PMID: 36610399; PMID: 18025878].

In addition, Lui et al. [PMID: 36610399] corroborates our study, suggesting an association between HERVK reactivation and inflammatory phenotypes, providing context and significance to our work. On the other hand, it is not surprising that we do not observe LINE-1 (L1) in the cytoplasm. Although L1s also have functional RT in humans, L1 retrotranscription occurs in the nucleus (unlike HERVK RT, which occurs in the cytosol).

These data provide strong evidence that the observed RNA-DNA hybrids signal in Figure 4 indeed originates from transposon expression, specifically HERVK activation. We have included the results of this experiment in the new Extended Data Figure 4, and is also reported here for Reviewer's convenience:

Finally, on line 245, the authors state cGas and STING are elevated in FAD cells via western blot, but the STING blot appears to be missing from the figure.

We apologize to the Reviewer for the oversight in the manuscript. There was a typo, the levels of STING were solely probed by immunofluorescence.

Regarding the claim of inflammation via the cGas/STING pathway, additional experiments are needed to show that the increase in cGas/STING protein expression is directly resulting in increased inflammation via this pathway. This can simply be shown via qPCR demonstrating elevated IFN expression in AD cells compared to controls. This will show that these cells are signaling for an inflammatory response. The authors could also utilize a commercially available cGAMP ELISA kit to show that cGas is binding to the hybrids and signaling towards STING activation.

We agree with the reviewer that our manuscript did not establish a direct link between STING protein levels and increased inflammation. However, our RNA-seq data (Figure 2b) reveal inflammatory pathway enrichment and INF- γ pathway activation in FAD progenitors, which also express higher levels of STING. A recent paper [PMID: 37239045] also supports this hypothesis, providing evidence that high levels of STING are associated with inflammation in the brains of patients diagnosed with a neurodegenerative disorder.

However, to provide a strong and conclusive link between STING protein levels and inflammation status, we conducted qPCR analysis to assess INF- γ levels, as suggested. Our results showed a significant increase in INF- γ levels in FAD lines compared to control, providing additional support for our conclusions. We have included these data in the revised Figure 5e and have also reported it here for the reviewer's convenience:

This manuscript provides strong evidence towards the correlation between early Alzheimer's disease and JUN upregulation and transposon activation. The addition of the recommended controls will solidify the transposon aspects of this research. We again thank the Reviewer for the positive feedback on our study. We are pleased to hear that the Reviewer finds our data to be strong evidence of the correlation between early Alzheimer's disease and JUN upregulation and transposon activation. We agree with their suggestion to solidify the transposon aspect of our research as shown in the above revisions, as they significantly strengthen our findings.

Reviewer #3 (Remarks to the Author):

Scopa et al propose an interesting hypothesis for the role of aberrant c-JUN expression in early AD pathogenesis, which impacts transposable element activation, neurogenesis and the immune response. To investigate this mechanism, they primarily use iPSC-derived hippocampal progenitor cells, and conduct a few additional studies in iPSC-derived neurons and organoids. While the premise is interesting and I appreciate the amount of work presented here, there are several major issues that give me cause for concern.

The majority of the manuscript is centered on 2xFAD vs 2xCTRL iPSC lines. Without an isogenic pairing or the use of multiple clones, this very small N means that it is very difficult to determine whether any difference between groups is actually due to FAD or due to the individual characteristics of each iPSC line (which are almost always of greater effect size than even an autosomal dominant mutation). The authors have identified many strikingly significant differences between groups throughout this work, however it does not appear that the correct statistical tests have been applied. The technical N has not been provided for any of the data presented - as an example Figure 1D, what is each data point? One well? One differentiation? What is each large dot? Presumably an average of each differentiation? I could not find this information for any figure throughout the whole manuscript. This brings me to the question about statistical tests - the methods describe an N=3 and a student's t-test used for analyses. This is not appropriate for the data - the authors have here an N=2 with multiple technical replicates. The stats as described are hugely inflating the significance of any test by essentially replicating data points and not taking into account that they are not independent observations.

A more appropriate test to use while maintaining some power and variability would be a linear mixed model. Similarly, I could not find the N you used for your RNA-seq anywhere, but an equivalent approach, e.g. DREAM would be required to take account of technical replicates (which differentiations are) to avoid inflating effect sizes. My second major concern is also related to the small N=2, with regards to their differentiation propensity. In the first figure the authors describe how CTRL and FAD lines differentiate into hpNPCs differently, with FAD lines seemingly generating more intermediate progenitors and fewer early neural stem cells. I do not think the authors have sufficient evidence to demonstrate that these differences are genuinely due to some pathogenic mechanism in FAD lines, rather than technical variation and differentiation capacity between different iPSC lines. iPSC lines and clones are notorious for highly variable propensities for differentiation into different cell types, and resulting cells have very variable levels of marker gene expression. Based on the data presented, the lack of isogenic controls and the small N, I don't think we can be confident that this is specific to FAD lines. More work needs to be done to dig into this further - e.g. in the absence of isogenic/more lines: utilize the RNA-seq data to dig more into cellular identity/PCA, try different differentiation protocols, direct neuronal induction, sequence the iPSCs etc.

Unfortunately, the concerns in point no.2 colour interpretation of the rest of the manuscript. If the cellular populations are so different between CTRL and FAD, how can you justify comparing the two on a bulk level by RNA-seq? How many gene expression changes are due to FAD and how many are due to different cell types? e.g. WNT and JUN expression are essential for neuronal differentiation - do you see higher expression of JUN because the FAD lines are better at differentiating and are more mature than the CTRLs? Similarly for differentiation pathways found in the RNA-seq - is it a disease-associated mechanism or reflective of different cellular populations being sequenced? This extends into the neuronal and organoid data - it is not surprising that neuronal populations would differ given that the NPCs are not the same. A generic neuronal marker would be helpful for Fig.2 to determine if you are seeing fewer neurons generally (and more glia) or specifically fewer CA3 neurons in the FAD lines. Was the subsequent RNAseq corrected for different CA3 proportion? How much is disease phenotype and how much is stemming from iPSC line differentiation capacity? The authors point out MAPT as a gene of interest spanning all pathways, however MAPT is also a useful generic neuronal marker, and again could be popping up due to differences in cell type proportions between cultures, rather than being differentially expressed in FAD lines. Differences in differentiation capacity and maturity may also underlie the ATAC-seq and TE data - it has been reported that there is TE de-repression during neuronal differentiation and substantial remodeling of chromatin - are the differences in these things due to an AD pathogenic mechanism, or technical artefact in differentiation capacity? Overall, the manuscript and hypothesis are very interesting, and may well be correct. However, given the experimental design and statistical analyses used, at this point the conclusions are difficult to justify given the nature of the iPSC line characteristics, differentiation differences and N.

The reviewer raises three main concerns about the study:

1. The small number of iPSC lines used, especially given the lack of isogenic controls.
2. Unclear definition of N in some of the plots and used incorrect statistical tests for some of the comparisons.
3. Not sufficient evidence to demonstrate that the differences observed are genuinely due to AD pathology, rather than technical variation and differentiation capacity between different iPSC lines.

Regarding the first concern:

To address the concern about the lack of isogenic lines, we have added an isogenic FAD pair (*PSEN2*:p.Asn141Ile with its isogenic control). We have repeated the main experiments of the

study on the isogenic pair, and **the results of these experiments replicated all the main findings previously reported with the other lines.**

Specifically, we found that the isogenic FAD line shows:

- Increased JUN expression in the FAD progenitors, both at gene and protein level.
- Significant increase of cytosolic RNA-DNA hybrids in the FAD progenitors.
- cGAS/STING activation in the FAD progenitors.

These results support that the findings previously reported in the first iteration of the manuscript were not an artifact of the genomic background of the FAD (and SAD) lines but instead due to the shared AD pathology. All the experiments and analyses conducted on the isogenic pair are reported in the revised Extended Data Figure 4.

We also remark that even in the first version of the manuscript, **all major novel findings were replicated not just in the two FAD lines, but also in the two SAD lines.** These replicated findings include the presence of TE-derived cytosolic RNA-DNA hybrids, the increase of cGAS/STING levels, and, most importantly, the reduced activation of the TE-RNA-DNA hybrid-STING axis upon c-JUN inhibition.

Since all these findings were replicated across four different AD lines: two FAD (with two different AD mutations) and two SAD sub-types, in addition to the newly incorporated isogenic pair, we strongly believe that the sample size or the genetic background of the lines did not generate significant technical artifacts.

Regarding the second concern:

We apologize for the absence of a detailed explanation of the SuperPlots presented throughout the manuscript. We have added a description of this type of graph in the Materials and Methods section of the revised manuscript, and here for the reviewer's convenience:

“A SuperPlot is a type of graph used to visualize individual data points and their averages. The distinct combinations of colors and shapes indicate the three separate experiments performed. Each small dot in the graph corresponds to a specific data point representing an analyzed image or cells. The larger dots represent the average values calculated from the respective data points”.

As already specified in the Materials and Methods section of the original manuscript, all the experiments (qPCR, immunofluorescences, and western blots) were repeated at least three times as independent experiments, with each experiment performed in technical duplicate or triplicate, as is standard practice in the field.

Finally, we apologize for the oversight in not clarifying the N for the NGS data. For each NGS analysis (RNA-seq and ATAC-seq) for each of the 6 lines (2 CTRL, 2 FAD, 2 SAD) we performed two replicates, corresponding to two different rounds of differentiation.

We understand Reviewer-3's concern that our previous statistical analysis of the RNA-seq data did not take into account the technical replicates, which could have inflated the significance of the results. To address this concern, we re-analyzed the data using a specific function of DESeq2, which considers the technical replicates as co-variates. This allows for the addition of the technical replicate as a co-variate, also adjusting for batch effects in the differential expression analysis by first fitting a linear model to the data. We found that re-analyzing the data with this approach did not affect the magnitude or significance of the results. For example, the number of differentially expressed (DE) genes previously obtained in the CTRL vs FAD progenitors' comparison (1,976) is comparable to the number of DE genes obtained when technical replicates are considered using this batch feature (1,973). Moreover, the results of the pathway analysis and upstream regulator analysis did not change.

The reviewer was also concerned about the statistical test used to analyze the immunofluorescence data. They recommended to employ a linear mixed model which takes technical replicates into account. Therefore, we employed a “Repeated Measures ANOVA test”, which is a particular type of mixed model specifically designed for this type of analysis. This type of ANOVA provides a more accurate estimate of the difference between the two conditions (CTRL and FAD) considering the technical replicates of each experiment.

Specifically, the Repeated Measures ANOVA test takes into account the correlation between the data points within each condition.

Importantly, we re-analyzed all the immunofluorescence throughout the paper using this Repeated measures ANOVA test. This did not affect the findings, as the p-values were confirmed to be significant in all the comparison previously identified as significant by the original t-test.

Regarding the third concern:

Neural progenitor cells (NPCs) are a highly heterogeneous population whose main activities and characteristics, such as quiescence/survival, proliferation, migration, differentiation, and integration (the different steps of neurogenesis), are regulated by intrinsic and extrinsic signals in the microenvironment [PMID: 34440814]. In vitro, the composition of the NPC population is therefore dependent on the culture conditions.

We would like to underline that in our study, we maintained NPCs in a proliferative state (the medium contained bFGF) for at least 21 days after differentiation before performing any experiment. This allowed the NPCs to show their own neurogenic features.

We agree with Reviewer-3 that the genetic background and epigenetic modifications among the iPSC lines may contribute to possible differences in the differentiation capability of the hippocampal NPCs (hpNPCs). However, we observed consistent trends in differentiation patterns between CTRL and FAD by repeating the differentiation several times and observing an increase in the intermediate hpNPC population in each FAD line. This suggests that while cell line variability exists, there are meaningful differences attributable to disease-specific mechanisms (as also now demonstrated in the isogenic lines).

This difference in hpNPC population composition and neurogenesis is in line with the literature in the field. Extensive evidence supports the notion that hippocampal neurogenesis is compromised in Alzheimer's disease (AD) both in human and animal models [PMID: 21323664, PMID: 30911133]. Recent studies indicate a decrease in immature neurons and stem cells in AD brains, suggesting the possibility of using adult neurogenesis as a cognitive functional measure in AD patients [PMID: 31130513; PMID: 32418723]. Of equal importance, there is mounting evidence proposing that alterations in hippocampal neurogenesis frequently precedes cognitive deficits and other hallmarks of AD [PMID: 31719242]. Numerous studies indicate that promoting healthy hippocampal neurogenesis, or inhibiting abnormal neurogenesis, may improve cognitive function and mitigate deficits in AD. Over the past few years, studies using transgenic models of AD have generated mounting evidence supporting a strong causal link between AD-associated genes and proteins and impaired neurogenesis [PMID: 22192775, PMID: 35503338]. These findings suggest that targeting neurogenesis may be a promising therapeutic strategy for AD.

Finally, we agree with Reviewer-3 that differential expression may be influenced by the difference in the composition of the hpNPC population between CTRL and FAD lines.

However, based on evidence in the literature as mentioned above, and our use of multiple AD cell lines, we can conclude that neurogenesis is impaired in FAD lines compared to CTRL due to a disease-specific mechanism rather than differences in their genetic background. In this perspective, it is necessary to analyze the entire heterogeneous pool of hpNPCs.

We applied this principle equally to our investigation of the CA3 neuronal population, which, in the context of our in-vitro study, represents the ultimate outcome of the process of impaired neurogenesis.

Nonetheless, upon discussion with the Editor, we have added a paragraph in the discussion which highlights that the RNA-seq results must be interpreted with caution due to the heterogeneity of the progenitor populations.

Other comments:

1. There are many places throughout the manuscript where there is some over-interpretation in the text, e.g. where is the evidence for AP-1 and BAF recruitment in these data? Where is

demonstration that activating cGAS-STING induces caspase 3 activation? These things may be associated in the literature, but be careful to not state you have demonstrated them in

Regarding AP-1 recruitment, the inhibition of the AP-1 subunit, c-JUN, leads to a significant decrease in transposable element transcription (see Figure 7a) as well a near complete elimination of TE-derived RNA-DNA hybrids (see Figure 7a). This would not be possible if c-JUN (AP-1) was not directly regulating these sites. The recruitment of BAF by AP-1 has been demonstrated by many studies in the literature [PMID: 29272704; PMID: 31375262]. However, we agree that some of our conclusions in the original version of the manuscript may have been overstated in the absence of direct experiments. Therefore, in the revised version of the manuscript, we have moderated such over-interpretations, in particular those referring to the AP-1/BAF link.

Finally, as the reviewer reported, several studies demonstrate that the cGAS-STING pathway promotes cell death by increasing the production of the pro-apoptotic protein BAX, which mediates caspase 3 activation [PMID: 30935414; PMID: 32253733 ;PMID: 34718659].

To address this point directly, we treated FAD progenitors with a well-established STING inhibitor (H151 compound). Notably, STING inhibition led to a significant reduction of cleaved caspase-3 levels. We believe that these findings provide clear evidence that the cGAS-STING pathway can promote apoptosis in FAD progenitors, and we have incorporated it into the revised manuscript in the revised Extended Data Figure 3d. We also reported the data here for the convenience of the reviewer.

2. Validation of your apoptosis hypothesis leading from caspase activation should be straightforward to do with IF. Do you think the different cell populations resulting from differentiation are due to increased cell death or altered differentiation? Or both?

Thank you for your comment. We understand the importance of IF to investigate the relationship between caspase activation and the different cell types. However, we would like to emphasize that the focus of our study is on understanding the collective disease-specific neurogenic behavior of the heterogeneous population of hpNPCs between CTRL and FAD samples. Therefore, we believe that a single-cell approach for this type of analysis, such as IF, is out of the scope of this study.

3. Given that you identify STING and its involvement in interferon responses, is it worth going back to your very first IPA analysis where interferon signaling was the most significant pathway? What's the relevance of it?

We apologize to the Reviewer, but we are not sure we have understood this specific comment. As they report, the pathway analysis in Figure 2b shows the interferon signaling and its response pathway as enriched in FAD progenitors. This data is consistent with our findings regarding the activation of the cGAS-STING pathway in the same samples. Furthermore, in the revised version of the manuscript we performed qPCR analysis to assess interferon gamma levels, as suggested. Our results showed a significant increase in interferon gamma levels in FAD samples compared to the control group.

4. Around figure 4, you stop presenting both CTRL lines - are both still being used? Why not include that data?

All the control and FAD lines were used in all the experiments and analyses. However, due to space limitations, we decided to show only one representative image for each condition. An exception was made for the immunofluorescence panels in Figure 4 and Figure 5. In this case, we show both FAD lines to demonstrate that the cytoplasmic accumulation of RNA-DNA hybrids and the consequent upregulation of STING (the main findings of the manuscript) are line/mutation-independent.

5. It would be useful to co-stain STING/RNA-DNA hybrid images with specific population markers, to see if they are specific to more mature progenitor populations

We appreciate the reviewer's comment. We believe that we have already demonstrated in the original version of the manuscript, particularly in the revised Figure 8 in the organoid context, that the RNA-DNA hybrid/STING signature is present in both progenitors (TBR2+) and in neurons (MAP2+).

Nonetheless, to further address this specific concern, we conducted a comprehensive co-immunostaining analysis on the three populations of CTRL and FAD hpNPCs, including the isogenic lines, using Nestin/Sox2 (for early progenitors) and DCX (for neuroblasts). This analysis was performed to assess the presence of JUN, RNA-DNA hybrids, and STING to support the novel axis described in the manuscript. The immunofluorescence demonstrated a consistent overexpression of c-JUN, RNA-DNA hybrids, and STING in all three lines of FAD hpNPCs (including the isogenic), regardless of cell type. We have incorporated these results into the revised manuscript in the revised Extended Data Figure 4. We also report the data here for the convenience of the reviewer.

6. Figure 5, referring to these data as "rescue" of the IP pool feels misleading, as you are reducing their number (or at least expression of the markers) rather than rescuing them. There is also even lower Nestin expression and little change in your other markers - so what is actually happening? What are the cells if not increasing the stem cell progenitor pool? Related to major concerns - is it really surprising that inhibiting c-JUN, a regulator of differentiation, reduces differentiation?

Thank you for this valuable comment. We appreciate the concern about the terminology used to describe the effects observed in Figure 5. The reviewer is correct that the term "rescue" may not accurately reflect the reduction in the number or expression of certain markers in the IP pool. We apologize for any confusion this may have caused. We changed the terminology in the revised version of the manuscript.

We expected that the FAD-induced impairment of neurogenesis is not simply due to the overexpression and upregulation of c-JUN. Therefore, we did not expect to see a complete "rescue" of all neurogenic features. It is important to remember that the dynamics of neural progenitor cell populations are complex and multifaceted. In fact, various extracellular and intracellular stimuli have been shown to modulate the survival, proliferation, and differentiation of NPCs in the hippocampus [PMID: 18786562, PMID: 21609825, PMID: 29922131]. For example, it is known that Sox2 is highly expressed in early progenitor cells and regulates their proliferative capacity and their potential to differentiate into multiple cell types [PMID: 26430216]. In our hpNPCs, Sox2 is downregulated in FAD, even after c-JUN inhibition (FAD vs. CTRL FC = -2.11; FADinhib vs. CTRL FC = -2.26). This might suggest that the reduction in Nestin expression is likely due to the depletion of the early progenitor pool.

c-JUN regulates neuronal differentiation, as Reviewer 3 noted, but this does not necessarily mean that it promotes an increase in the number of differentiated cells. In fact, c-JUN has been shown to influence also the proliferation of neural progenitor cells in the hippocampus and to inhibit neuronal differentiation [PMID: 19262166, PMID: 16491125].

Given the intricate interplay of multifaceted factors regulating impaired neurogenesis, it is not surprising that the inhibition of c-JUN is not sufficient to achieve a complete "rescue" of the observed phenomena in FAD hpNPCs.

7. Make sure to check references are correct - 86 in the methods should be 87. I did not go through and check others. How do Lancaster organoids relate to hpNPCs and hippocampus? How did you determine what "proper" neuronal differentiation was? The FADs and SADs actually have much nicer looking rosettes than controls, and may be better at making neurons (consistent with all your other data). The IF data here is also odd (particularly in controls) - I wouldn't expect SOX2 to have any overlap with NeuN and CTIP2. We thank the reviewer for pointing out the reference error in the methods section. We have corrected the reference number as suggested and checked all the other references and citations.

Lancaster's organoids protocol produces organoids comprising multiple regions corresponding to various components of the brain, such as dorsal and ventral forebrain, hindbrain, and hippocampus. As the authors reported: 'After one month, organoids should begin to exhibit neuronal differentiation, marked by Tuj1 or DCX leading to progressive expansion and thickening of cerebral tissues over the subsequent 1-2 months. At this stage, a number of different brain regions are visible... hippocampus marked by Prox1 and Fzd9 staining' [PMID: 25188634].

We believe that Lancaster's organoids provide a three-dimensional neural tissue model that can recapitulate neurogenesis in a more complex in-vitro system than the standard two-dimensional, single-type culture. We determined "proper" neuronal differentiation by using a combination of morphological and molecular criteria, as described in the Lancaster protocol. These criteria include the expression of neuron-specific markers (sox2, Tju1 (data not shown), NeuN, Cipt2) and the morphology and organization of the cells (presence of Sox2+ rosettes and arborization of neurons).

We apologize for the CTRL organoids panel in Figure 8a. It is true that typically in-vivo, SOX2 expression does not co-localize with mature neuron markers such as C1TP2. However, the heterogeneity and nuances of in-vitro differentiation can sometimes produce less distinct results. As the Reviewer can see in the revised Extended Data Figure 8, we have provided an improved version of the same image. This revised image shows that the expression of Sox2 is lower in the rosette than in the other rosettes where C1TP2 is not co-expressed. We apologize for this oversight, which we now recognize was caused by an error in the image uploading and processing procedures used to create the panel.

8. Overall, what is your proposed mechanism for upregulated JUN and dysregulated chromatin accessibility in FAD and SAD lines? They presumably have no pathology yet, MAPT expression is low, is it Ab processing? Is it a stress response - to what?

We thank the reviewer for this comment. At present, the exact upstream trigger leading to the upregulation of c-JUN remains undetermined. We are planning and initiating supplementary investigations that delve into the underlying mechanisms.

REVIEWERS' COMMENTS

Reviewer #1 (Remarks to the Author):

Scopa et al. have made a nice revision. This reviewer has no more question.

Reviewer #2 (Remarks to the Author):

The authors addressed all of my concerns.

Reviewer #3 (Remarks to the Author):

The authors have made some helpful edits and clarifications to their manuscript, and I appreciate the amount of work required to replicate some of these experiments in an additional iPSC line.

I still have some outstanding questions and comments:

1. The most major concern is still the lack of N or statistical information throughout the manuscript. The authors confirm the N for their NGS in their response to reviewers document, but I did not see this in the manuscript (apologies if I missed it). I also appreciate and understand the use of superplots to summarize quantification data, but again there was no N information. Is each dot one cell, one image, one z-plane, one field of view, one well etc? There are also western blots and IF images throughout without quantification with no information regarding reproducibility. The N ought to be explicit in every figure legend (e.g. N=2, 3 independent differentiations), not just in the statistical information ("at least 3 replicates") every time.

2. Related to this, It's unclear why the heatmap in figure 3e should have 4 control columns and 3 FAD columns when based on 2 lines each?

3. Minor comments: It would be helpful to have a sentence or two regarding the other upstream DE gene regulators WNT5A, TNF and MAPT - and why they weren't followed up for analysis when they were more significant than JUN,

4. Are all of the images in Figure 8A the same power? The third panel looks lower, or did the treatment change cell morphologies and make them smaller?

5. I think you have the images in ED 4C the wrong way round - the control/FAD look like they ought to be on the left side of the panel, not the top. One DAPI image also appears to be a c-jun overlay, not DAPI alone

6. Typo line 417 (studyiong)

Dear Editor,

Thank you very much for accepting in principle our manuscript (NCOMMS-22-50575A). Here are our answers to the residual reviewer comments.

Reviewer #3 (Remarks to the Author):

The authors have made some helpful edits and clarifications to their manuscript, and I appreciate the amount of work required to replicate some of these experiments in an additional iPSC line.

I still have some outstanding questions and comments:

1. The most major concern is still the lack of N or statistical information throughout the manuscript. The authors confirm the N for their NGS in their response to reviewers document, but I did not see this in the manuscript (apologies if I missed it). I also appreciate and understand the use of superplots to summarize quantification data, but again there was no N information. Is each dot one cell, one image, one z-plane, one field of view, one well etc? There are also western blots and IF images throughout without quantification with no information regarding reproducibility. The N ought to be explicit in every figure legend (e.g. N=2, 3 independent differentiations), not just in the statistical information ("at least 3 replicates") every time.

We have now added the required information explicitly in every figure legend.

2. Related to this, It's unclear why the heatmap in figure 3e should have 4 control columns and 3 FAD columns when based on 2 lines each?

As explained in the figure legend, one of the four FAD RNA-seq samples was discarded for poor data quality.

3. Minor comments: It would be helpful to have a sentence or two regarding the other upstream DE gene regulators WNT5A, TNF and MAPT - and why they weren't followed up for analysis when they were more significant than JUN.

As requested, we have added a sentence at the end of the discussion to explain why only JUN was followed-up among the upstream regulators.

4. Are all of the images in Figure 8A the same power? The third panel looks lower, or did the treatment change cell morphologies and make them smaller?

Yes, all the images in Fig. 8A have the same power and the treatment did NOT make changes to cell morphology.

5. I think you have the images in ED 4C the wrong way round - the control/FAD look like they ought to be on the left side of the panel, not the top. One DAPI image also appears to be a c-jun overlay, not DAPI alone

The reviewer is correct, the figure panel is now fixed accordingly.

6. Typo line 417 (studyiong)

Fixed.